# The anti-angiogenic and anti-proliferative activity of n-benzoyl-3-phenyl isoserine derivative containing 1,2,3-triazole and quinoline rings

Nawar Raad Hussein[ID][1,2*], Hayder B. Sahib[3], Zahraa Sabbar Omran[4]

1 College of Pharmacy, Al-Farahidi University, Baghdad, Iraq, 2 Department of Pharmacology and Toxicology, College of Medicine, AL-Nahrain University, Iraq, 3 Department of Pharmacology and Toxicology, College of Pharmacy, AL-Nahrain University, Baghdad, Iraq, 4 Department of Biochemistry, College of Medicine, Kerbala University, Karbala, Iraq

* ph.nawar@gmail.com; bpharmacist8@gmail.com

## Abstract

One of the biggest causes of death globally is cancer, which develops tumors characterized by angiogenesis and excessive cell proliferation. Cancer treatments targeting these pathways, especially those involving vascular endothelial growth factor, have had encouraging results. This study examined the anti-angiogenic and anti-proliferative properties of a phenyl isoserine derivative compound. The *ex vivo* anti-angiogenic activity was examined by rat aorta ring (RAR) and *in vivo* chorioallantoic membrane (CAM) assays. The *in vitro* anti-proliferative properties of the compound were analyzed utilizing Human umbilical vein endothelial (HUVEC), colon cancer, and breast cancer cell lines. Quantitative real-time PCR was used to measure gene expression of vascular endothelial growth factor receptor 2 (*VEGFR-2*) gene. An evaluation was conducted on the compound's SwissADME profile, molecular docking, and molecular dynamics simulations against the VEGFR-2 receptor. The phenyl isoserine derivative demonstrated significant anti-angiogenic activity in both the RAR and CAM assays, with a half-maximal inhibitory Concentration ($IC_{50}$) value of 18.91 µg/mL in the RAR assay. The compound also exhibited anti-proliferative effects against HUVEC, colon cancer, and breast cancer cells, with $IC_{50}$ values of 68.85, 124.75, and 112.31 µg/mL, respectively. The results of the Real-Time PCR test showed that at a concentration of 200 µg/mL, there was a significant decrease in the expression of the *VEGFR-2* gene in Kaposi's Sarcoma (KS) cells. The compound exhibited membrane permeability and dispersion, as expected based on the SwissADME profile, with a consensus *log P $_{o/w}$* value of 3.14, while also indicating potential cytochrome P450 enzyme inhibition and poor water solubility (*Log $_{Sw}$* value: −8.34). Molecular docking (MD) studies revealed a higher PLP fitness score (91.496) for the compound compared to axitinib (78.748), suggesting a favorable binding to VEGFR-2. However, axitinib demonstrated a more negative binding energy (−14.548 kcal/mol vs. −7.715 kcal/mol), reflecting stronger energetic stabilization under docking

**Data availability statement:** All relevant data are within the paper.

**Funding:** The author(s) received no specific funding for this work.

**Competing interests:** The authors have declared that no competing interests exist.

conditions. Molecular dynamics simulations showed that ligand exhibits stable binding and induces greater structural stabilization of the target protein compared to axitinib, suggesting its potential as a more effective VEGFR-2 inhibitor. The phenyl isoserine derivative was found to have a potent inhibitory effect on angiogenesis and *VEGFR-2* gene activity, indicating it could be a promising candidate for anti-cancer medicine.

## Introduction

Angiogenesis is a critical event for tumor growth, as transitioning from a dormant tumor to metastatic expansion requires the formation of new capillaries derived from surrounding vessels. Blood vessels (BV) originate from blood-forming stem cells and serve as conduits for disseminating hematopoietic cells throughout the organism, particularly in diseases such as leukemia [1]. Significant interest has been directed toward cytokine receptor inhibitors, especially those targeting the vascular endothelial growth factor (VEGF) and its receptor VEGFR. By directly inhibiting VEGF, these compounds prevent signal transduction and deactivate the signaling cascade of the VEGFR receptor, effectively blocking the stimulatory receptor [2].

The cytokine network associated with pro-inflammatory effects includes interleukin-1 beta (IL-1β), IL-6, and tumor necrosis factor-alpha (TNF-α). The secretion of pro-inflammatory IL-1β is activated by various stimuli, such as cell damage and lipopolysaccharides (LPS) derived from bacterial membranes. IL-6, a pleiotropic pro-inflammatory cytokine, promotes acute-phase reactions, hematopoiesis, and specific immunological responses [3]. Additionally, inhibitors of receptor tyrosine kinases, such as epidermal growth factor (EGF), fibroblast growth factor (FGF), and platelet-derived growth factor (PDGF), block their respective growth factor receptors [2,3].

VEGFR-2 inhibitors often include a consistent pharmacophore model that includes four critical components: a hinge-binding heteroaromatic ring, a hydrophobic central scaffold, interaction with the aspartate-phenylalanine-glycine (DFG) motif, and a solvent-exposed polar tail. These characteristics facilitate high-affinity and selective inhibition via critical interactions with hinge residues (e.g., Cys919), the hydrophobic pocket, and Asp1046 within the DFG motif. [4]. Many FDA-approved inhibitors of the VEGFR-2 enzyme (axitinib, cabozantinib, lenvatinib, sorafenib, and tivozanib) possess four critical pharmacophoric characteristics: a hinge-binding heteroaromatic ring, a central aromatic linker, a pharmacophoric hydrogen bond donor/acceptor moiety interacting with the DFG motif, and a terminal lipophilic group that occupies the allosteric pocket. Previous studies have effectively integrated these characteristics into quinoxaline-based frameworks, showcasing their promise as VEGFR-2-targeted anticancer drugs [5].

The 1,2,3-triazole moiety can mimic numerous functional groups and act as a bioisostere in various bioactive molecules within pharmaceutical chemistry [6]. Compounds containing 1,2,3-triazole and diaryl ether have demonstrated significant anti-tumor efficacy by inhibiting Bax/Bcl-xL linkage proteins in cancer cells [7,8]. Furthermore, 1,2,3-triazole conjugates have been evaluated for their anti-proliferative

effects against liver cancer cell lines (HepG2) and breast cancer cell lines (MCF-7), with ellipticine, a natural topoisomerase II inhibitor, serving as a positive control. These compounds significantly inhibited HepG2 and MCF-7 cancer cell proliferation by over 30% at low concentrations (<10 μM) [9].

The biological importance of the 1,2,3-triazole moiety extends to the synthesis of potent compounds with diverse activities, including antioxidant, anti-inflammatory, antibacterial, and anticancer properties. This versatility is attributed to the ability of 1,2,3-triazoles to interact with various enzymes and receptors through distinct bonding mechanisms. Notably, 1,4-dihydropyridine-based triazole derivatives exhibit cytotoxic activity against the Caco-2 (colorectal adenocarcinoma) cell line, underscoring their potential as viable candidates for anti-cancer drug development [10].

The quinoline scaffold is appealing due to its pharmacological properties, including anti-HIV, antineoplastic, antiasthmatic, antituberculosis, antifungal, and antibacterial activities [11]. These properties stem from the presence of a bicyclic heteroaromatic pharmacophore within the quinoline structure [12]. During pharmacophore development using the scaffold-hopping strategy, extensively documented methodologies, alongside their corresponding biological activities, have been evaluated. Notably, the antiproliferative activity of 2-styrylquinoline-3-carboxylate derivatives is particularly significant, especially when phenyl and carboxylate groups are attached at the C-2 and C-3 positions of the quinoline scaffold [13–15].

The biological activity of certain 1,2,3-triazole derivatives is being investigated to explore their significant medicinal potential. The metabolic stability, lack of toxicity, and favorable physical characteristics of the 1,2,3-triazole core make these compounds strong candidates for therapeutic development. Many biologically active 1,2,3-triazoles have been documented in the literature, including the 1,2,3-triazole motif in medications, which are increasing annually. These compounds play a critical role in formulations exhibiting notable biological activities, including antibacterial, antifungal, antiviral, antituberculosis, anti-Alzheimer's, antimalarial, anti-inflammatory, and anti-cancer effects, among others [16].

The prediction of pharmacokinetic parameters serves as a preliminary phase in preclinical drug trials. Lipinski and his team pioneered an *in-silico* approach for predicting pharmacokinetic parameters (absorption, distribution, metabolism, and excretion (ADME)) [17]. Predicting ADME parameters has proven essential in reducing the likelihood of failure during drug candidate development [18].

This study investigated the anti-angiogenic and anti-proliferative activity of N-benzoyl-3-phenyl isoserine derivatives containing 1,2,3-triazole. These activities were evaluated using the aortic ring assay, chick embryo chorioallantoic membrane assay, antiproliferative assays for three different cell lines: human umbilical vascular endothelial cells (HUVECs), human colon cancer cell lines (HCT-116), and human breast cancer cells (MCF-7), gene expression analysis through real-time quantitative polymerase chain reaction (RT-qPCR), molecular docking, molecular dynamics simulations and ADME estimation for pharmacokinetic parameter prediction.

## Materials and Methods

This study was conducted at Al-Nahrain University, College of Medicine, Department of Pharmacology, from December 1, 2022, to July 1, 2023. All experimental procedures adhered to the College of Medicine/Al-Nahrain University Committee for Animal Care standards and received approval from the Research Ethical Committee of the College of Medicine/Al-Nahrain University (clearance No: UNCO-MIRB20240510). The novel chemical compound investigated in this study was referred to as the "N2", and its structure is presented in (Fig 1). Its chemical structure is 3-benzamido-3-phenyl-2-(4-((quinoline-8-yloxy) methyl)-1H-1,2,3-triazol-1-yl) propanoic acid ($C_{28}H_{23}N_5O_4$). This compound exhibits anti-inflammatory and anti-cytokine storm properties. The compound was provided by the College of Medicine, Al-Nahrain University, Department of Biochemistry, and synthesized according to the method described by Sabbar Omran Z. [19].

The synthesis of the N2 and its intermediates involved dissolving N-benzoyl-(2R,3S)-2-phenyl isoserine in dimethylformamide (DMF) with triethylamine (TEA), followed by the addition of p-toluenesulfonyl chloride. The mixture was heated at

**Fig 1. Chemical structure of compound N2.** (N2: 3-benzamido-3-phenyl-2-(4-((quinoline-8-yloxy) (methyl))-1H- 1,2,3-triazole-1-yl) propanoic acid) (N-Benzoyl-3-phenyl isoserine derivative containing 1,2,3-triazole and quinoline rings).

40°C for 4 hours, then cooled, filtered, and washed to yield 3-benzamido-3-phenyl-2-(tosyloxy)propanoic acid. This product was then dissolved in DMF, and sodium azide was added. The mixture was heated at 60°C for 7 hours, after which the solvent was evaporated and the residue filtered to obtain 2-azido-3-benzamido-3-phenylpropanoic acid. Next, 3-hydroxyquinoline was combined with acetone and potassium carbonate. The mixture was heated to 50°C for 7 hours, then cooled and filtered to yield (3-ethyloxy) quinoline. The 2-azido-3-benzamido-3-phenylpropanoic acid, CuSO4·5H2O, and sodium ascorbate were dissolved in methanol, and a solution of (3-ethyloxy) quinoline was introduced. The mixture was heated at 40°C for 60 hours and then cooled to yield the N2 [19].

## Aortic ring assay

Rat aortic ring explant cultures were established according to Madri and Williams's method with minor modifications [20,21]. Sprague Dawley rats were housed under conventional living conditions for the one-week acclimatization period, with a 12-hour light/dark cycle, and they had unrestricted access to clean water and standard feed pellets. To ensure deep anesthesia and the absence of responsiveness to pain, animals were given intraperitoneal injections of ketamine (80 mg/kg) and xylazine (10 mg/kg) before the aortic ring experiments. Aortic rings were obtained from six male Sprague-Dawley rats. The aortas were dissected into 1 mm cross-sections, thoroughly washed with serum-free medium, and cleared of fibro-adipose tissue remnants. The Experiment was conducted in 48-well tissue culture plates (Costar Corning, USA). Each well contained 500 µL of 3 mg/mL fibrinogen in serum-free M199 growth medium, supplemented with 5 µg/mL aprotinin to inhibit fibrinolysis. A tissue piece was positioned at the center of each well, and 10 µL of thrombin (50 NIH U/mL) in 0.15 M NaCl was added. After embedding the vessel segment in fibrin gels, 0.5 mL of M199 medium, enriched with 20% heat-inactivated fetal bovine serum (HIFBS), 0.1% ε-aminocaproic acid, 1% L-glutamine, and 0.6% gentamicin, was introduced to each well. N2 was included in the growth medium at concentrations ranging from 6.25 µg/mL to 200 µg/mL. Negative control cultures received media containing Dimethyl sulfoxide (DMSO) vehicle purchased from Sigma-Aldrich

(Germany). Bevacizumab, a prominent anti-angiogenic drug, was purchased from Sigma-Aldrich (Germany) and utilized as a positive control. Cultures were maintained at 37°C in 5% $CO_2$ for 5 days.

The extent of blood vessel outgrowth on day five was measured using an inverted microscope (Olympus, Japan) equipped with a digital camera (Leica CCD, Japan) and Leica Qin computerized imaging software. The inhibition of vessel growth was assessed as the mean percentage of inhibition relative to the negative control ± SD (standard deviation). The degree of blockage of blood vessel growth was evaluated using the Nicosia method [22].

### Dose-response study with ex vivo aorta ring assay

Serial dilutions of the analyzed sample were prepared at the following concentrations: 200, 100, 50, 25, 12.5, and 6.25 μg/mL. Samples were solubilized in DMSO (10 mg of test agent in 1 mL DMSO) and then diluted in M199 growth medium to achieve a final DMSO concentration of 1%. Wells without test samples were treated with a medium containing 1% DMSO as the negative control. The data were presented as mean ± standard deviation (SD). 100 μg/ml Bevacizumab was used as a positive control. The concentration inhibiting 50% of proliferating blood vessels, referred to as "IC50," was determined using either the linear regression or the semilogarithmic equation for the extract. Let Y represent the percentage of inhibition and X denote the concentration [23].

### Chick embryo chorioallantoic membrane assay

The in vivo anti-angiogenic action of the N2 was examined using the chick chorioallantoic membrane (CAM) test. Fertilized chicken eggs obtained from a local hatchery in Baghdad were disinfected with 70% ethanol and then incubated for approximately 72 hours at 37°C with a relative humidity of 60%. The fertilized eggs were placed in an incubator immediately after the onset of embryogenesis, maintained at a constant humidity and temperature of 37°C. On day 3, a square aperture was created in the shell after extracting 2 mL of albumen to separate the CAM from the shell. The window was secured with clear adhesive tape, and incubation continued until the day of the experiment, on day 4 of incubation [24,25]. The CAM formed a flat membrane at the top, extending to the edge of the dish, creating a two-dimensional monolayer for the placement of the N2. One milligram of the N2 was placed into each 6 mm of Whitman filter paper and subsequently dried at 45–50°C. Discs containing the vehicle (1% DMSO) served as the negative control, while one milligram of Bevacizumab was employed as the positive control. The loaded and dried discs were inverted and placed on the CAM [25]. Images of each CAM were captured using a digital camera, and the dimensions of each CAM were recorded using the SketchAndCalc application (an irregular area calculator app for images containing irregular shapes) [26]. The responses were classified as + (3–6 mm), ++ (6–9 mm), and +++ (≥ 10 mm) as a score scale—the measurement of the inhibitory zone [27].

### Cell lines and growth conditions

The American Type Culture Collection (ATCC) supplied the human umbilical vein endothelial cell line (HUVEC), human colon cancer cell line (HCT-116), and human breast cancer cell line (MCF-7). Human umbilical vein endothelial cells (HUVECs) were cultured in Dulbecco's Modified Eagle's Medium (DMEM) supplemented with 10% fetal calf serum (FCS), 10 ng/mL vascular endothelial growth factor (VEGF), and antibiotics. HUVECs were used up to the seventh passage. Human cancer cells were cultured at 37°C with 5% $CO_2$ in 50 cm2 flasks (Nunc Brand, Denmark) containing 5 mL of Roswell Park Memorial Institute – 1640 Medium (RPMI-1640) (Gibco, USA), supplemented with 10% fetal bovine serum and antibiotics ($10^3$ IU penicillin and 0.001 g streptomycin). Human cancer cells were used for a maximum of four passages. Furthermore, 0.5 mL of trypsin (Nutricell, Brazil) was added to the culture flasks to dissociate the adhering cell lines. Trypsin was subsequently inactivated by adding 5 mL of RPMI 1640 supplemented with 5% fetal bovine serum. Cells were carefully pipetted to promote their separation in the fluid. Before plating the cells onto culture plates, they were

enumerated and diluted to the appropriate seeding densities. The substances were evaluated at a concentration of 100 μg/mL in triplicate wells. The cells were treated with a DMSO medium as a negative control [28].

## Antiproliferative Assay

The MTT assay, utilizing 3-(4, 5-dimethylthiazol-2-yl)-2, 5-diphenyltetrazolium bromide, was employed to assess cell line growth following the Mosmann method [29]. HUVECs and human cancer cells were seeded in 96-well culture plates at a density of $1 \times 10^4$ cells per well. MTT was prepared by dissolving 5 mg/mL in phosphate-buffered saline (PBS). Twenty microliters of MTT were used per well, and the plates were incubated at 37°C in 5% $CO_2$ for five hours [30]. Cells at 70–80% confluence were treated with N2 samples at concentrations ranging from 6.25 to 200 μg/mL and cultured at 37°C in 5% $CO_2$ for 48 hours. The plates were removed from the incubator, and the supernatant was aspirated. Two hundred microliters of DMSO were added to each well. The plates were vigorously agitated for one minute at ambient temperature to dissolve the dark blue crystals. The absorbance was measured at 570 nm, with the reference at 650 nm, using a micro-plate reader (HumaReader HS). Cells cultivated in control media were considered to reflect viability 100%. The viability of treated cells was assessed as a percentage of the untreated control's viability. Each concentration was evaluated in triplicate. The proportion of cell line inhibition was calculated as the mean ± standard deviation using the following equation:

Cell inhibition (%) = 1- {(A0/A) × 100}

(A0 = Absorbance of sample and A = Absorbance of control)

The $IC_{50}$, representing the inhibitory effect on 50% of cell growth, was determined to assess the growth inhibition profile of the N2. The linear and logarithmic correlation equations were used to compute the $IC_{50}$ values [29].

## Gene expression by real-time quantitative polymerase chain reaction (qPCR) for VEGF

On the day before treatment, Kaposi's sarcoma cancer cells were cultured at $1 \times 10^4$ cells per flask. The medium was replaced with a fresh formulation containing six concentrations of the N2: 12.5, 25, 50, 100, and 200 μg/mL. Bevacizumab was employed as the positive control. The cells were incubated for 16 hours at 37°C and 5% CO2 in two distinct hypoxic environments (0.1% $O_2$). Hypoxic conditions were maintained using a Gas Pak Pouch (Becton Dickinson, Sparks, MD, USA). The cells were then collected for RNA extraction. The medium was aspirated, and total RNA was extracted using Trizol reagent for gene expression analysis. RNA samples were treated with RNase-free DNase (Promega, USA) and then subjected to ethanol precipitation. The purity of total RNA was assessed via spectrophotometry, followed by reverse transcription using Superscript II (Promega, USA) and an oligo-dT20 primer. One microgram of total RNA was converted to complementary DNA (cDNA) in a final volume of 20 μL using the PCR process for 1 hour [31].

The gene examined in this experiment was the *VEGFR-2* gene [32]. VEGF Quanti-Tect SYBR Green primers were obtained from Qiagen, Germany. The *VEGFR-2* gene primers were CCCACTGAGGAGTCCAACAT (forward) and CAT-ACCTCCCCTGTGCAACT (reverse), while the glyceraldehyde-3-phosphate dehydrogenase (GAPDH) gene, serving as a housekeeping gene, had primers GTCTCCTCTGACTTCAACAGCG (forward) and ACCACCCTGTTGCTGTAGCCAA (reverse). This study utilized GAPDH primers from Invitrogen-USA [33].

The IQ4 real-time quantitative PCR apparatus (Bio-Rad, USA) and the SYBR Green kit were employed to detect total mRNA expression. This was conducted to assess the comparative mRNA expression of VEGF in the Kaposi's sarcoma cell line. The reaction mixture consisted of 12.5 μL of Go-Taq qPCR Master Mix (Promega, USA), 2.0 μL of complementary DNA, 2.5 μL of primers, and 8 μL of DNase-free distilled water, totaling 25 μL. PCR amplification was performed using an iQ5 Cycler (Bio-Rad) under the following conditions: 95°C for 3 minutes, followed by 40 cycles of denaturation at 95°C for 15 seconds, annealing at 60°C for 45 seconds, and an extension step at 72°C for 20 seconds. The plate was maintained at 80°C for 15 seconds. A melting curve was conducted for each compound to determine the specificity of each PCR primer by reducing the temperature to 55°C and then elevating it to 95°C at a rate of 0.5°C per 10 seconds. Gene expression quantification was determined using the standard curve and cycle threshold for each quantity of the N2. The

gene expression levels were standardized to the reference gene expression, and the fold change was calculated relative to the untreated cell control. The findings were normalized using the housekeeping gene (GAPDH) by the delta-delta cycle threshold (ΔΔCT) method. Data analysis was conducted by employing the relative quantitation methodology, defined as the ratio of the reference gene GAPDH to the gene of interest (GOI), as described by the Livak method [34].

## ADME/Tox profile

The in silico ADME/Tox profile is an effective tool for predicting the pharmacological and toxicological characteristics of drug candidates, particularly during the preclinical phases. In silico models have been utilized to enhance ADME/Tox predictions [35]. The SwissADME web tool (http://www.swissadme.ch/) offers accessible computational methods for a comprehensive evaluation of the pharmacokinetic profiles of small compounds. The web tool developers selected methodologies for their robustness and interpretability to enable practical application in medicinal chemistry. The online tool developers modified some of these methodologies using open-source methods, while others remained unchanged replicas of the original techniques [36]. The molecular structures of the N2 were presented using simplified molecular-input line-entry specification (SMILES) nomenclature (O=C(c1ccccc1) NC(C(n1nnc(c1) COc1ccc2c(c1) cccn2) C(=O) O) c1ccccc1) to the ADME/Tox web tools pkCSM and SwissADME for pharmacokinetics and pharmacodynamics analysis. We identified the significant ADME/Tox features from the web resources to characterize the ADME/Tox profile.

## Molecular docking studies

**Measurement of Precise Linear Potential (PLP) fitness score.** Molecular docking analyses were conducted using the Cambridge Crystallographic Data Centre (CCDC) GOLD 2022 software. The target proteins were acquired from the Protein Data Bank (PDB). The human vascular endothelial growth factor receptor 2 (VEGFR2) Kinase Domain complexed with AAL993 (PDB: 5EW3) served as the target for the synthesized compounds, with axitinib used as a reference medication. The docking method was executed following the official GOLD user guide published by the CCDC [37]. The synthesized compounds and the reference pharmaceuticals were rendered with Chem3D software. The energy was minimized using Avogadro software and the UFF force field as the minimization algorithm. The target protein was imported into the Hermes visualizer, a component of the CCDC-Discovery suite. Hydrogens were incorporated into the protein, and the active site residues were examined for tautomerism and ionization states. Superfluous chains, ligands, water molecules, and cofactors were removed. The protein's original ligand was removed. The protein's active site was occupied by the reference drug and the final compounds, which produced docking solutions. A region of 10 Å was designated as the pocket for the active site. The docked molecules were quantified using the PLP (Precise Linear Potential) fitness score. The intermolecular interactions of each docked molecule were visualized and documented [38].

## Measurement of the Binding energy

Docking is a computer technique that forecasts the optimal orientation of one molecule relative to another when they associate to create a stable complex. Docking has been extensively employed to propose the binding mechanisms of protein inhibitors [39]. The Glide software from the Schrödinger suite (Schrödinger Maestro, version 140132) was used for molecular docking experiments. The sitting of Glide software: force field: Optimize Potentials for Liquid Simulations 4 (OPLS4), Solation model: Variant Specific Generalized Born/Surface Area (VSGB), simulation pH: 7.5, optimization uses: PROPKA (predict the protonation state of specific atoms within a protein or ligand at a particular pH). The Protein Data Bank website was employed to choose the protein, which was subsequently stored in PDB format [40]. This experiment focused on examining the interactions between the vascular endothelial growth factor receptor 2 (VEGFR2) (PDB: 5EW3), the N2, and the reference medicine axitinib. The Protein Data Bank obtained the receptor's crystallographic structure to study the protein interactions. Hydrogen atoms were added and removed from solvent molecules during preparation to

create a clean and suitable environment for docking simulations. The protein-ligand complex was reduced to confirm that the ligands fit well in the protein binding site. The co-crystallized binding ligand was used as a reference for molecular docking. To direct the docking simulations, this grid was helpful in identifying potential binding sites in the catalytic region of the target proteins [41].

## Molecular dynamics simulation

Molecular dynamics simulations were conducted for the complexes (axitinib and N2 with the target protein vascular endothelial growth factor receptor (PDB ID: 5EW3), and the protein alone (PDB ID: 5EW3)). The system was prepared using the Transferable Intermolecular Potential-3 Point (TIP3P) water model and simulated using GROMACS 2023.0 on Linux 22.4. The following setup includes Ligand and Protein Modeling. Ligand topologies were generated using the SwissParam server. The protein was modeled using the CHARMM27 force field. System Preparation: A salt concentration of 0.15 mol/L was added to ensure system neutrality. Energy minimization was performed using the steepest descent method for 50,000 steps. NVT (constant number of particles, volume, and temperature) and NPT (constant pressure and temperature) equilibration phases were performed at 310 K and 1 atm. V-rescale thermostat and Parrinello-Rahman barostat were employed to maintain temperature and pressure stability. The 100-nanosecond production run was conducted at 310 K. The following metrics were calculated to assess system stability and interactions: principal component analysis (PCA), Solvent-Accessible Surface Area (SASA), Root Mean Square Deviation (RMSD), Radius of Gyration (ROG), Root Mean Square Fluctuation (RMSF), and Hydrogen bonding (H-bonding) analysis. Plots for these analyses were generated using Xmgrace software, pymol, and the matplotlib Python library [42].

## Statistical analysis

The obtained data were expressed as the mean and standard deviation (SD). The differences between the means were considered significant at $P < 0.05$. A one-way analysis of variance (ANOVA) followed by a Tukey test comparison (2-tailed) was used to compare treatment groups. Non-linear regression analysis using GraphPad Prism software version 9.0 was employed to determine the IC50 (the concentration that causes 50% growth inhibition of cultured cells).

## Results

### Aortic ring assay

**Rat aorta ring anti-angiogenic ex vivo assay (RAR).** A 100 µg/mL concentration of each N2 and Bevacizumab (positive control) was administered to the rat aorta, which was then fixed in whole-growth medium (M199)(Fig 2) ((2: b), (2: g), and (2: h)) illustrates the activity of each compound on blood vessel growth in rat aorta. The inhibition of blood vessel growth was expressed as the mean percentage ± standard deviation (SD). The N2 and Bevacizumab significantly inhibited blood vessel growth by 99.33 ± 0.58% and 95.8 ± 1.3%, respectively, on day five. A significant difference was detected in blood vessel inhibition between the N2, Bevacizumab (positive control), and 1% DMSO (negative control) ($p < 0.05$) (Fig 3a)

### Dose-response relationship for the N2 on blood vessel inhibition in rat aorta ring assay (RAR assay)

Six concentrations of the N2 (200, 100, 50, 25, 12.5, and 6.25 µg/mL) are shown in (Fig 2: a,2: b, 2: c, 2: d, 2: e, and 2: f), respectively. These concentrations demonstrated significant dose-dependent inhibition activity ($p < 0.05$) compared to the negative control (1% DMSO). The inhibition percentages were characterized as mean ± SD as follows: 100%, 99% ± 0.58, 99% ± 0.58, 83% ± 3.05, 74% ± 4.5, 30% ± 4.04, and 14% ± 2.51 for the respective concentrations. Blood vessel growth was assessed five days after the experiment (n = 18). The $IC_{50}$ value was determined using a semilogarithmic equation,

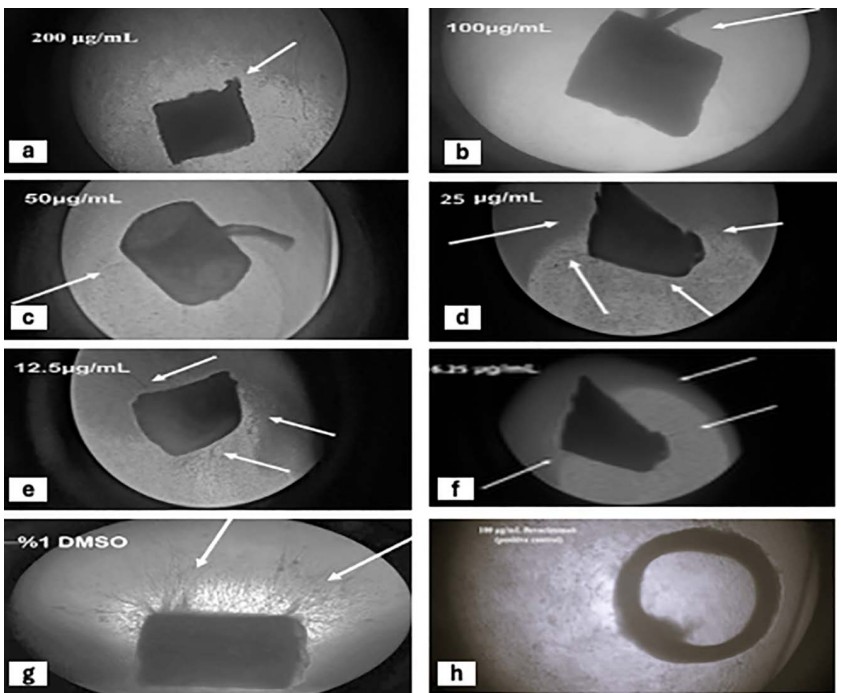

**Fig. 2. The impact of successive N2 levels: a: 200 µg/ml, b: 100 µg/ml, c: 50 µg/ml, d: 25 µg/ml, e: 12.5 µg/ml, f: 6.25 µg/ml, g: 1% DMSO (negative control), and h: 100 µg/ml Bevacizumab (positive control) on blood vessel formation in rat aorta rings.** The arrows indicate the development of micro-blood vessels. The images were taken on day five of the experiment.

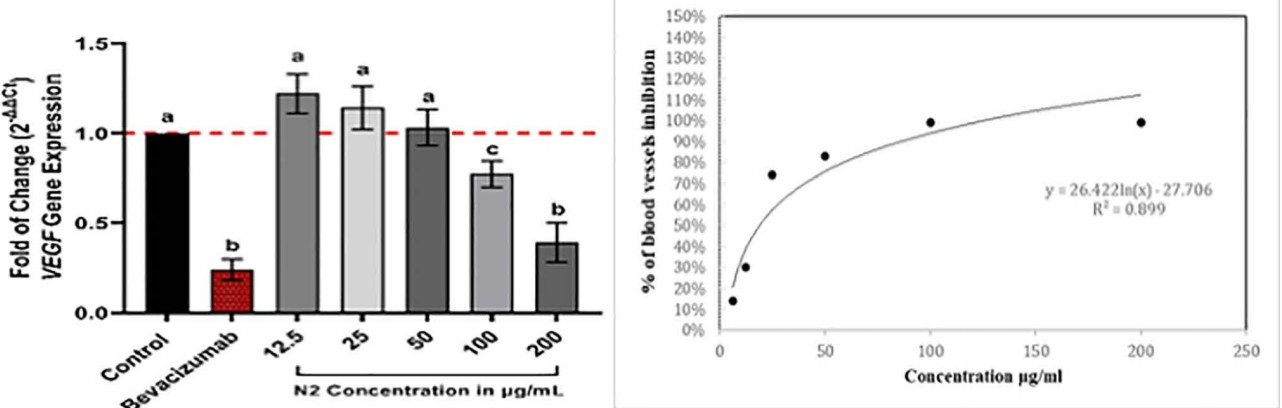

**Fig 3. Dose-response relationship for the N2: (a) Anti-angiogenesis activity of the differentN2 (12.5–200 µg/mL) (Mean ± SD percentage of blood vessels inhibition) of the N2 along with Bevacizumab (positive control) and 1% DMSO (negative control) using ANOVA post hoc (Tukey's test).** Different letters (a, b, c, d, e, f) were considered significantly different at $p < 0.05$ in *ex vivo* aortic ring assay. **(b)** Dose-response curve of the inhibitory effect of different N2 (12.5–200 µg/mL) on the rat aorta ring assay.

$Y = 26.422 \ln(x) - 27.706$, corresponding to 18.91 µg/mL, where Y represents the percentage of inhibition and X represents the concentration. The dose-response curve for the serial dilutions of the N2 applied to the rat aorta is shown in (Fig 3b).

## Chick chorioallantoic membrane assay (CAM assay)

The scoring system was used to assess the inhibition zones on day seven, and it was significant (+++) when blood vessel formation was inhibited by > 10 mm. This effect is presented in Table 1. In the chick chorioallantoic membrane assay (Fig 4), suppression of blood vessel growth is categorized into two groups: (A) a negative control CAM assay, (B) the N2 treatment groups, and (C) Bevacizumab (positive control) in the CAM assay. Table 1 and (Fig 4) show that the CAM treated with the N2 and Bevacizumab (positive control) exhibited a significant ($p < 0.05$) anti-angiogenic effect. Many vessels beneath the disc were inhibited, resulting in 1 mg of each compound.

## Effects of the N2 on HUVEC proliferation

The results demonstrated a dose-dependent suppression of cell proliferation after 48 hours. The N2 concentrations were 200, 100, 50, 25, 12.5, and 6.25 µg/mL, with each concentration tested in triplicate. The data were expressed as the mean ± standard deviation (SD). The inhibition percentages of HUVEC cell growth for the N2 at each specified concentration were 66 ± 14.84, 64 ± 14.57, 54 ± 7.77, 21 ± 2, 7 ± 1, and 0 ± 2.65, respectively. (Fig 5), illustrates the in vitro evaluation of the N2 on HUVEC cells at passage 7. The IC50 value for the N2, derived from the graph, was computed using a semi-logarithmic equation: $y = 22.011 \ln(x) - 43.147$, where Y represents the percentage of inhibition and X denotes concentration. The IC50 value for the N2 was 68.85 µg/mL.

**Table 1. The zone of inhibition of blood vessel growth and the corresponding scoring using the chick chorioallantoic membrane (CAM) assay, treated with the N2 and Bevacizumab (positive control) (n = 6 for each group).**

| NO. of Egg | Zone of inhibition area (mm) of N2 | Scoring1 | Zone of inhibition area (mm) of Bevacizumab | Scoring1 |
|---|---|---|---|---|
| 1 | 15 | +++ | 22 | +++ |
| 2 | 19 | +++ | 18 | +++ |
| 3 | 8 | + | 14 | +++ |
| 4 | 11 | +++ | 16 | +++ |
| 5 | 20 | +++ | 15 | +++ |
| 6 | 40 | +++ | 19 | +++ |

1(+++) indicates that blood vessel formation was inhibited by > 10 mm.

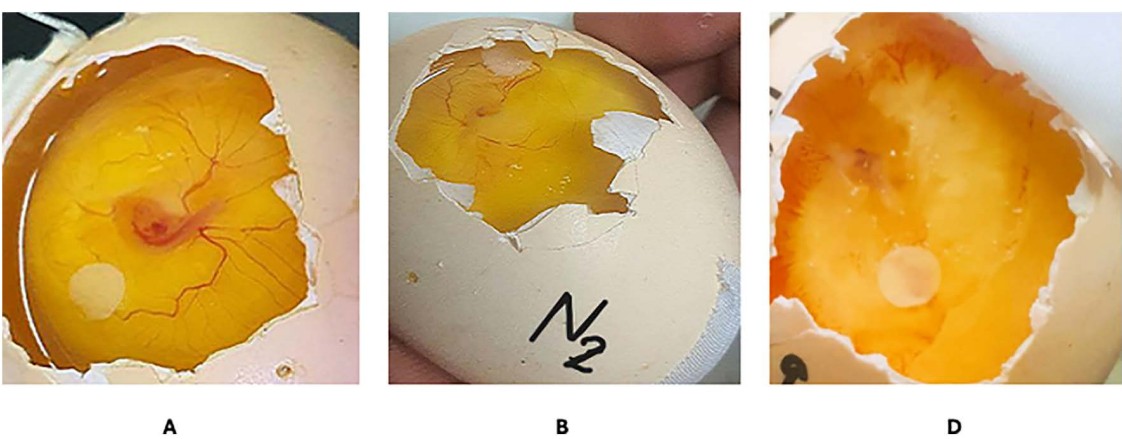

**A**  **B**  **D**

**Fig 4. The degree of the inhibition zone of blood vessel growth in vivo (CAM) assay: (A) 1% DMSO (negative control), (B) treated with N2, (C) Bevacizumab (positive control).**

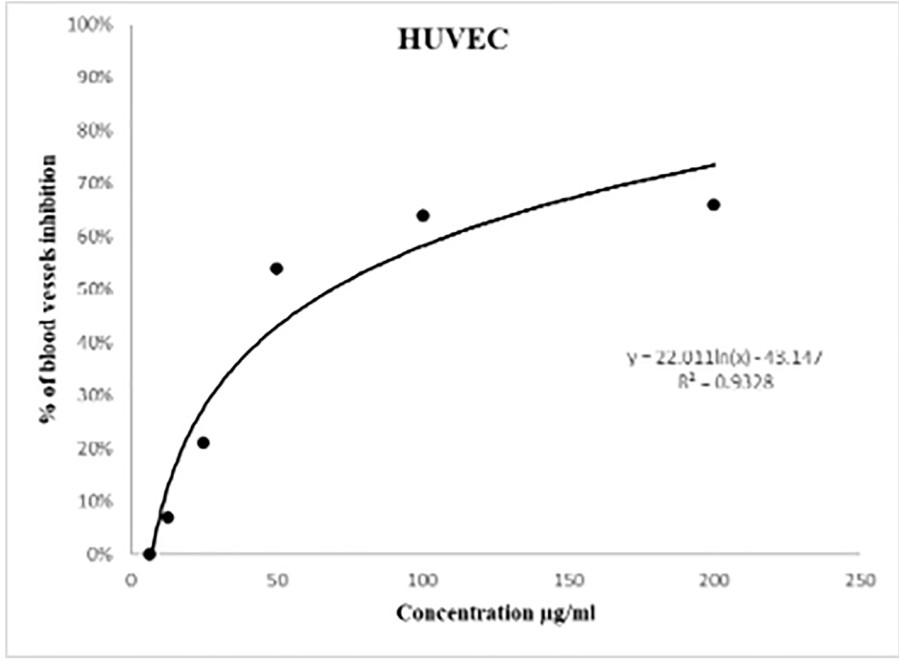

**Fig 5. The in vitro evaluation of the N2 on human umbilical vein endothelial cells (HUVEC).**

### Growth inhibition profile of human colon cancer (HCT116) and human breast cancer (MCF7) cells using the N2

The results showed that N2 exhibited cytotoxic properties against HCT116 and MCF7 cancer cell lines. (Fig 6), illustrates the in vitro evaluation of the N2 on HCT116 and MCF7 cells at passages 4–7. The results demonstrated a dose-dependent suppression of cellular proliferation after 48 hours. The percentages of HCT116 and MCF7 cell proliferation inhibition for the N2 at each concentration are shown in Table 2. The N2 demonstrated an inhibitory action with IC50 values of 124.75 and 112.31 μg/mL against HCT116 and MCF7 cells, respectively. The IC50 value is derived from the graph for the N2.

### Gene expression of vascular endothelial growth factors (VEGF) for Kaposi's sarcoma (KS) using Real-Time-PCR and the activity of the N2 on the *VEGFR-2* gene

Real-time PCR investigation revealed a down-regulation in *VEGFR-2* transcription levels in KS cell cultures, indicating that the N2 exhibited dose-dependent effectiveness. Fig 7 illustrates the dose-response effect of the N2 on *VEGFR-2* gene expression, showing a reduction in value relative to the control. Notably, KS cells were more significantly suppressed at the higher 200 μg/mL concentration than at the other concentrations. Furthermore, as shown in Table 3, the N2 dosages of 12.5, 25, 50, and 100 μg/mL resulted in *VEGFR-2* transcription reductions with P-values of 0.1175, 0.4874, 0.9983, and 0.1040, respectively, compared to the control values, which were non-significant in KS cells (P-value < 0.05). The 200 μg/mL concentration markedly decreased *VEGFR-2* transcription, with a $p < 0.0001$. This corresponds to roughly fold decreases for the final concentration treatments. Bevacizumab decreased *VEGFR-2* transcription, with a $p < 0.0001$,(Fig 7).

### Assessment of pharmacokinetics and the target proteins

Absorption was predicted based on water solubility and lipophilicity. Lipophilicity was assessed using the logarithm of the n-octanol/water partition coefficient, predicted using the consensus Log Po/w descriptor from SwissADME. Log Po/w is related to transport processes, including membrane permeability and distribution to various tissues and organs. For

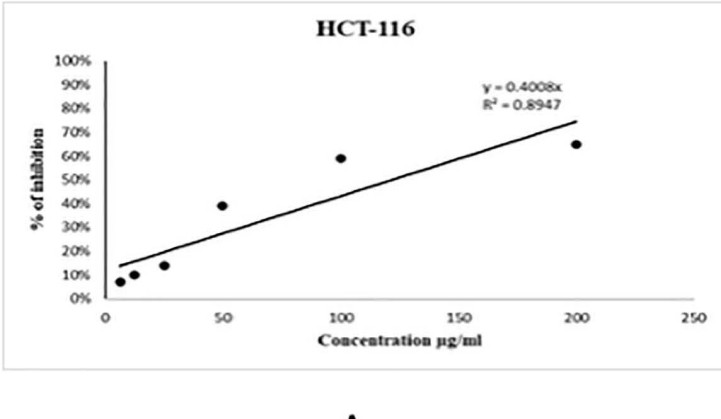
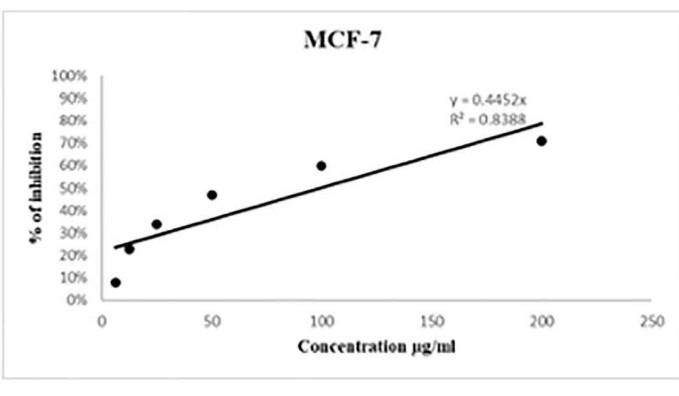

**A**                                                                                              **B**

**Fig 6. The in vitro screening of the N2 on human cancer cells: (A) Human colon cancer (HCT-116) and (B) Human breast cancer cell (MCF-7).**

**Table 2. Serial concentrations and their respective percentages (mean±SD) of inhibition of the N2 on human colon cancer (HCT116) and human breast cancer (MCF-7) cells.**

| Concentration (μg/mL) | Percentage of inhibition of HCT-116 (mean±SD) | Percentage of inhibition of MCF-7 (mean±SD) |
|---|---|---|
| 200 | 65±4.51 | 71±0.42 |
| 100 | 59±1.53 | 60±1.67 |
| 50 | 39±6.11 | 47±2.72 |
| 25 | 14±4.04 | 34±1.36 |
| 12.5 | 10±2.52 | 23±0.71 |
| 6.25 | 7±3.61 | 8±0.62 |

the N2, the predicted value of consensus $Log\ P_{o/w}$ was 3.14. Water solubility was estimated using the in-silico $IT\ Log_{Sw}$ descriptor from SwissADME. The expected Log Sw value for the chemical was −8.34. According to the SwissADME Log Sw scale, the N2, with values less than −6, is considered poorly soluble ($Log_{Sw}$ scale: insoluble<−10<poorly<−6<moderately<−4<soluble<−2<very<0<highly). All physicochemical, solubility, and pharmacokinetic results are illustrated in Table 4. Distribution was forecasted utilizing the glycoprotein P (*P-gp*) substrate and blood-brain barrier (BBB) permeability. Metabolism was assessed using SwissADME based on the inhibition of the principal cytochromes (CYP) of the P450 superfamily, specifically CYP3A4, CYP1A2, CYP2C9, CYP2D6, and CYP2C19. The inhibition of CYP enzymes, a primary mechanism for metabolism-related drug-drug interactions, typically involves competition between drugs for the same enzyme binding site. Drug-likeness descriptors chosen by Lipinski were computed using SwissADME. Excretion primarily involves renal and hepatic clearance and is associated with bioavailability.

The Swiss Target web application is designed to forecast small compounds' most relevant protein targets. The current analysis was confined to the top 15 Homo sapiens targets, as shown in (Fig 8). Regarding the prediction of potential target proteins by both entities, we selected targets with the greatest number of known actives (3D and 2D) as "d" for dimensions. A higher number of known actives (3D/2D) indicates an abundance of experimental data for that target protein, which helps evaluate the reliability of the predicted target and encourages further research. All targets listed in Table 5, have a ratio above 200/0 of known active compounds (3D/2D).

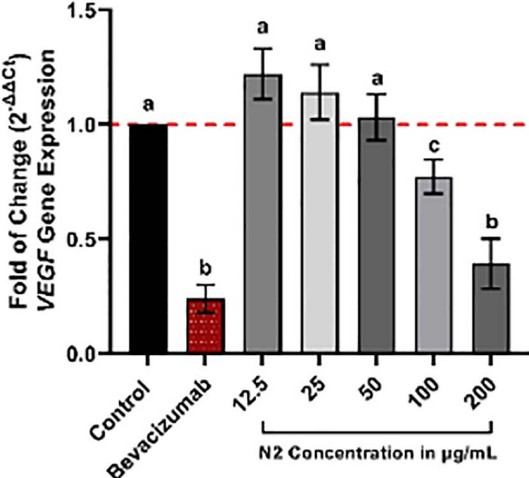

**Fig 7. Mean ±SD fold of *VEGFR-2* gene after exposure to different N2 concentrations (12.5–200 µg/mL) compared to Bevacizumab (positive vehicle) treatment.** Measurements of untreated cells were normalized to 1.0-fold expression and considered the control. Comparison between different groups was achieved using ANOVA post hoc (Tukey's test) and different letters (a, b, c) considered significantly different at p<0.05, determined by the ΔΔCt method.

**Table 3. Effect of different concentrations of N2 (n=18) on *VEGFR-2* gene expression.**

| Treatment N2 µg/mL | Mean±SD fold of change | Significant |
|---|---|---|
| Control | 1.0±0.0 | no |
| Bevacizumab | 0.27±0.10 | ** |
| 12.5 | 1.22±0.11 | no |
| 25 | 1.14±0.12 | no |
| 50 | 1.03±0.10 | no |
| 100 | 0.77±0.07 | no |
| 200 | 0.39±0.11 | ** |

**p<0.05. Bevacizumab (positive control)

## Molecular docking

The obtained docking scores are expressed by precision linearwise potential (PLP) fitness, which predicts the binding affinity of the docked compound with the human vascular endothelial growth factor receptor 2 (VEGFR-2) (PDB: 5EW3). The scoring function is based on PLP fitness, which models the steric and clashing interactions between the ligand and the protein. The N2 showed a PLP fitness of 91.496 and bonded with the target via two interactions: a Van Der Waals (VDW) interaction between the quinolone ring and valine 898, and a hydrogen bond between the triazole nitrogens and the nitrogen of the peptide linkage of aspartic acid 1046. In comparison, axitinib showed a PLP fitness score of 78.748 and demonstrated a hydrogen bond with the amino acid isoleucine 1025 via the amide nitrogen atom (Table 6), and (Fig 9).

## Molecular docking to measure Binding energy of the N2 with vascular endothelial growth factor receptor 2

Table 7, displays the relationships between the N2 and Axitinib (Reference drug) with the vascular endothelial growth factor receptor 2 (VEGFR-2) at the active site, the kinds of interactions, and the strength of the link between the residues and

**Table 4. ADME/Tox profile of the N2 predicted by SwissADME.**

| Parameters | Contents | Value |
|---|---|---|
|  | MW1 (g/mol) | 493.51 |
|  | No. heavy atom | 37 |
|  | No. aromatic heavy atoms | 27 |
| **Physicochemical properties** | No. rotatable bonds | 10 |
|  | No. H-bond acceptors | 7 |
|  | No. H-bond donors | 2 |
|  | Molar refractivity | 135.95 |
|  | TPSA2 | 119.23 |
| **Lipophilicity** | Consensus3 Log Po/w | 3.14 |
| **Water Solubility** |  | Poorly soluble |
| **Pharmacokinetics** | GI4 absorption | High |
|  | BBB5 permeant | No |
|  | P-gp6 substrate | Yes |
|  | CYP1A2* inhibitor | No |
|  | CYP2C19* inhibitor | No |
|  | CYP2C9* inhibitor | Yes |
|  | CYP2D6* inhibitor | Yes |
|  | CYP3A4* inhibitor | Yes |
|  | Log Kp | −6.69 |
| **Druglikeness** | Lipinski | Yes |
|  | Bioavailability | 0.55 |

1MW.: molecular weight, 2TPSA: topological polar surface area (Å²), 3Log Po/w: partition coefficient between n-octanol and water, 4GI absorption: gastrointestinal absorption, 5BBB permeant: blood-brain barrier permeation, 6P-gp: permeability glycoprotein, *: five main isoforms of cytochromes P450 (CYP1A2, CYP2C19, CYP2C9, CYP2D6, and CYP3A4), and 7Log Kp: skin permeability coefficient (cm/s).

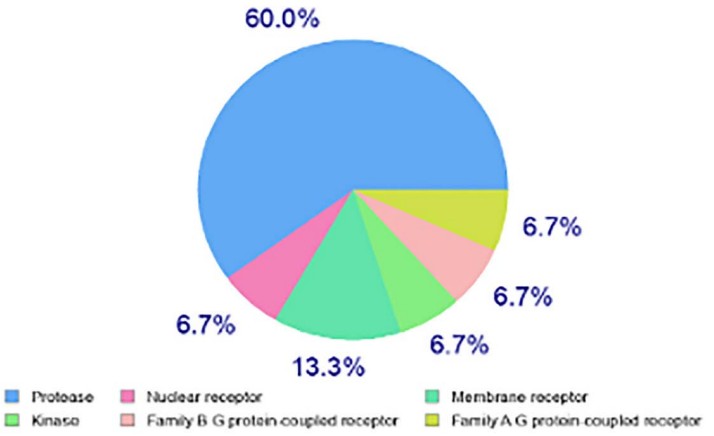

**Fig 8. Prediction of Swiss targets for the N2.**

the compound and shows the 2D and 3D interaction map of the N2 and Axitinib with human vascular endothelial growth factor receptor 2 (VEGFR-2) (Fig 10). More favorable energy stabilization inside the binding pocket is shown by Axitinib's much greater binding affinity (−14.548 kcal/mol) compared to N2 (−7.715 kcal/mol).

**Table 5. Pharmacodynamics of the discovered N2 as determined by Swiss Target Prediction.**

| Protein target | Target class | CHEMBL1 ID | Actives (3D/2D) |
|---|---|---|---|
| Peroxisome proliferator-activated receptor alpha | Nuclear receptor | CHEMBL239 | 486 / 0 |
| G protein-coupled receptor 44 | Family A G protein-coupled receptor | CHEMBL5071 | 344 / 0 |
| Integrin alpha-4/beta-7 | Membrane receptor | CHEMBL2095184 | 267 / 0 |
| Integrin alpha-V/beta-3 | Membrane receptor | CHEMBL1907598 | 243 / 5 |
| Angiotensin-converting enzyme | Protease | CHEMBL1808 | 230 / 0 |
| Neprilysin | Protease | CHEMBL1944 | 226 / 0 |

1CHEMBL ID: an identification from a chemical database containing bioactive compounds with pharmacological characteristics.

**Table 6. Ligand-protein interactions between VEGFR-2 (PDB: 5EW3) and N2, which has two interactions: one between the quinolone ring and valine (VAL) 898 through Van Der Waals (VDW) and another by a hydrogen bond (HB) between triazole nitrogens and the nitrogen of the peptide linkage of aspartic acid (ASP) 1046. Axitinib has one interaction with amino acid isoleucine (ILE) 1025 via the amide nitrogen atom.**

| Compound | PLP fitness | Interactions |
|---|---|---|
| N2 | 91.496 | ASP 1046 (HB*), VAL 848 (VDW*) |
| Axitinib | 78.748 | ILE 1025 (HB*) |

*HB: Hydrogen bond, VDW: Van Der Waals.

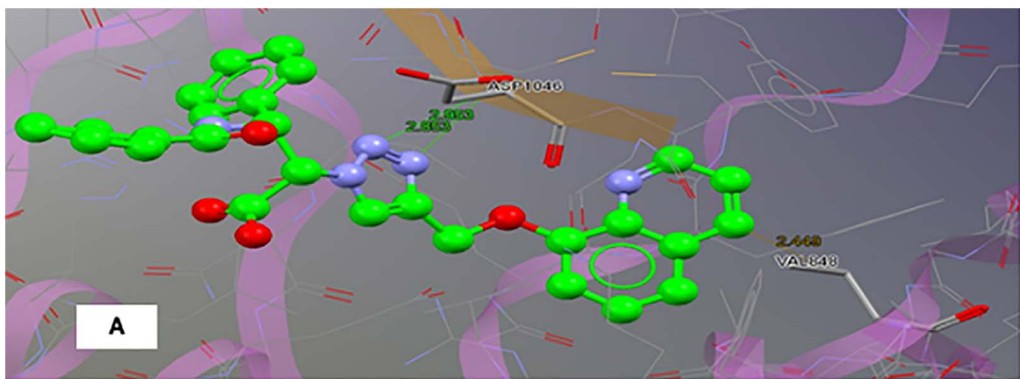

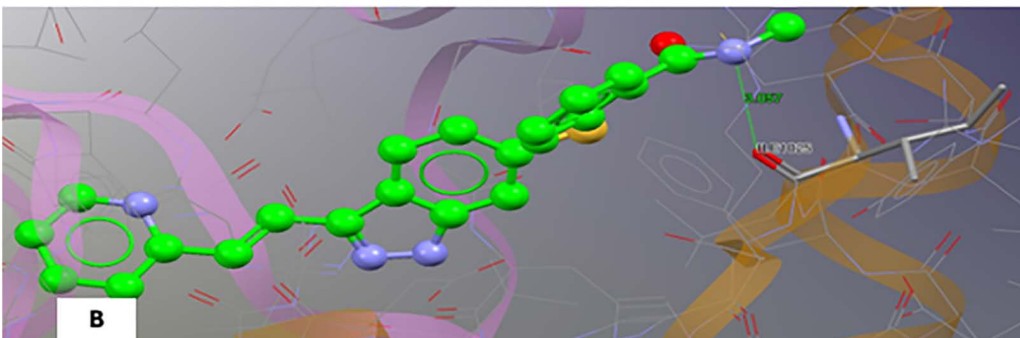

**Fig 9. 3D configuration for: (a) the N2 and (b) Axitinib.**

**Table 7. The strength of the link between the N2 and Axitinib with receptor (5EW3) of vascular endothelial growth factor receptor 2 (VEGFR-2).**

|  | Binding energy (kcal/mol) |
|---|---|
| N2 | −14.548 |
| Axitinib | −7.715 |

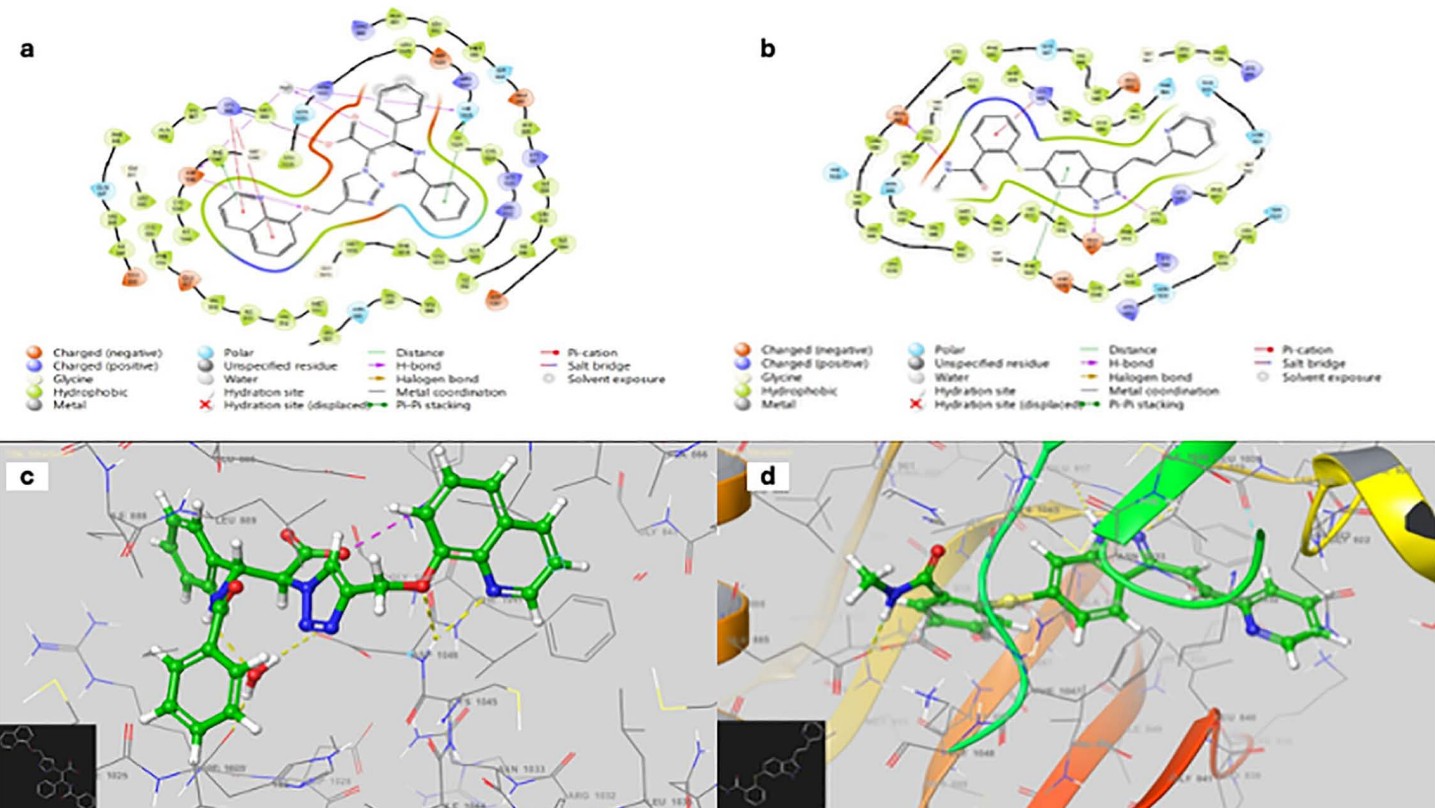

**Fig 10. The 2D and 3D interaction maps of the N2 and Axitinib with the receptor (5EW3) of vascular endothelial growth factor receptor 2 (VEGFR-2). (a) 2D interaction map of N2, (b) 2D interaction map of Axitinib, (c) 3D interaction map of N2 and (d) 3D interaction map of Axitinib.**

## Molecular dynamics simulations

**Interaction Analysis.** Interaction profiling provided additional insights into ligand behavior, revealing critical stabilizing contacts within the complexes. The FDA-approved (axitinib) protein complex maintained persistent hydrogen bonds, particularly with residues GLU65 and LYS48, along with stabilizing hydrophobic interactions with CYS169. These contacts contributed to the overall structural coherence of the complex and explained the moderate yet stable RMSD and Radius of Gyration ROG observations (Fig 11a). On the other hand, the N2 relied primarily on hydrogen bonds with LYS48, GLU65 and contacts with CYS169 and LEU159 by hydrophobic interactions (Fig 11 b).

**Principal Component Analysis (PCA).** PCA was applied to deconstruct the essential modes of motion within each system, capturing the dominant collective movements across the simulation. The apoprotein showed high conformational variability, with one principal component (PC1) accounting for 62.6% of the total motion and the cumulative contribution

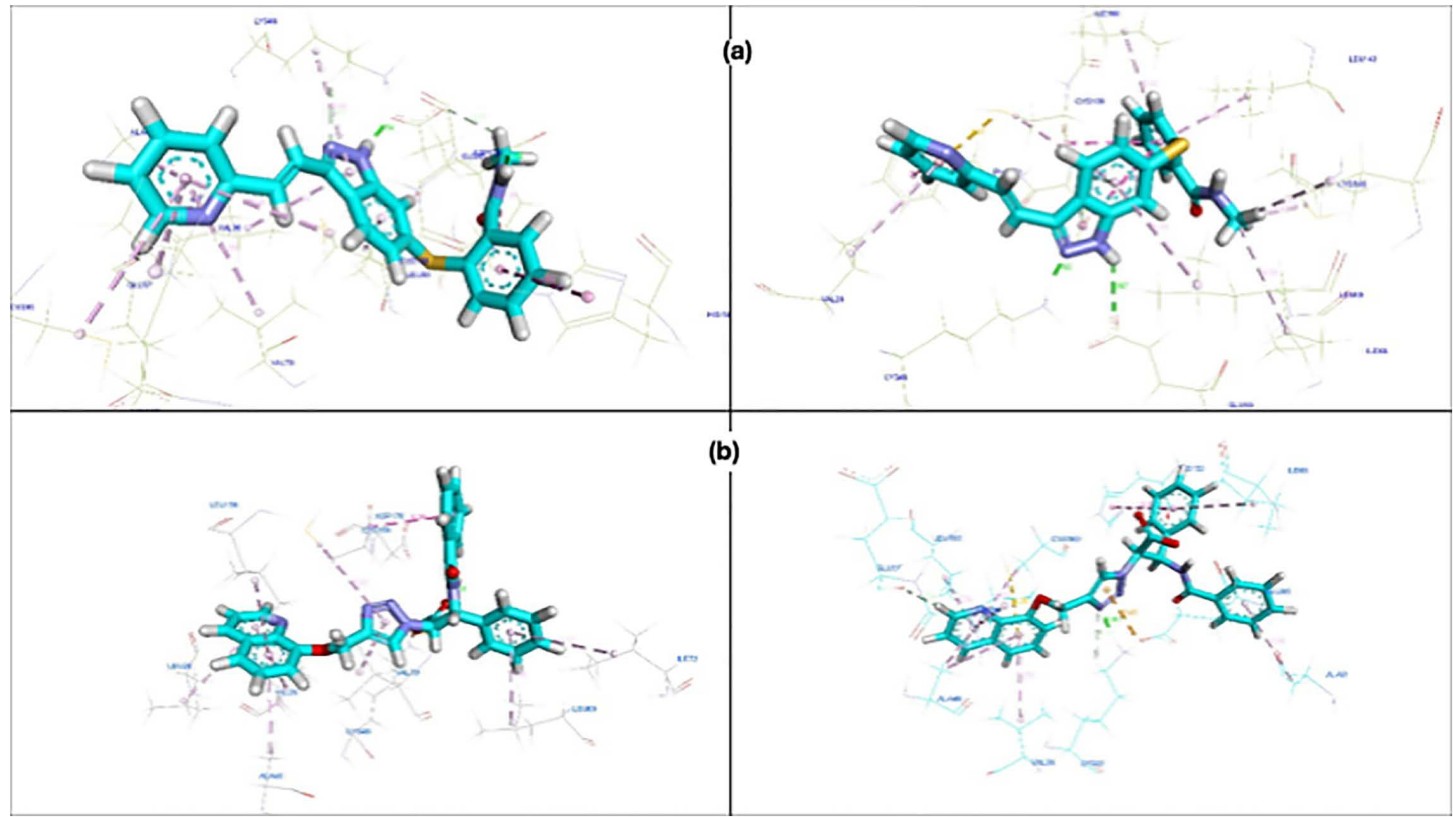

**Fig 11.** **(a) The first frame and last frame interaction of (5EW3) with Axitinib. (b) The first frame and last frame interaction of (5EW3) with N2.**

reaching 85% by the tenth principal component (PC10). This wide distribution underscores the inherent flexibility of the unbound protein, which is often essential for its functional repertoire. For the N2, PCA revealed a starkly different profile. The motion was heavily concentrated in the first few principal components, with PC1 alone capturing 79.8% of the dynamics and a cumulative contribution of 92% by PC10. This skewed distribution indicates that the system is dominated by limited, unidirectional movements consistent with a rigidified protein conformation. Such restriction could stem from the ligand's overly tight binding, which may hinder the necessary conformational adaptability of the protein. In comparison, Axitinib complex demonstrated a more balanced PCA profile. PC1 accounted for 68.0% of the total motion, and PC10's cumulative contribution reached 87%. These values reflect moderated flexibility, moderate restriction in protein movement, and loss of protein function (Fig 12).

### Solvent-Accessible Surface Area (SASA)

The area exposed to solvent molecules is measured by Solvent-Accessible Surface Area (SASA). Protein alone (about 150 nm²), with Axitinib (about 155 nm²), and with the N2 (155–160 nm²) are the three plots that display SASA with time. While Axitinib and N2 raise the average SASA, suggesting a small structural expansion, the protein alone exhibits dynamic changes. N2 has a comparable effect on the protein as Axitinib, albeit there may be some minor differences (Fig 13).

### Root Mean Square Deviation (RMSD)

Across all simulations, the protein undergoes minor conformational changes upon ligand binding, stabilizing RMSD values after the initial equilibration phase. Axitinib and the N2 exhibit low RMSD fluctuations, indicating high structural stability

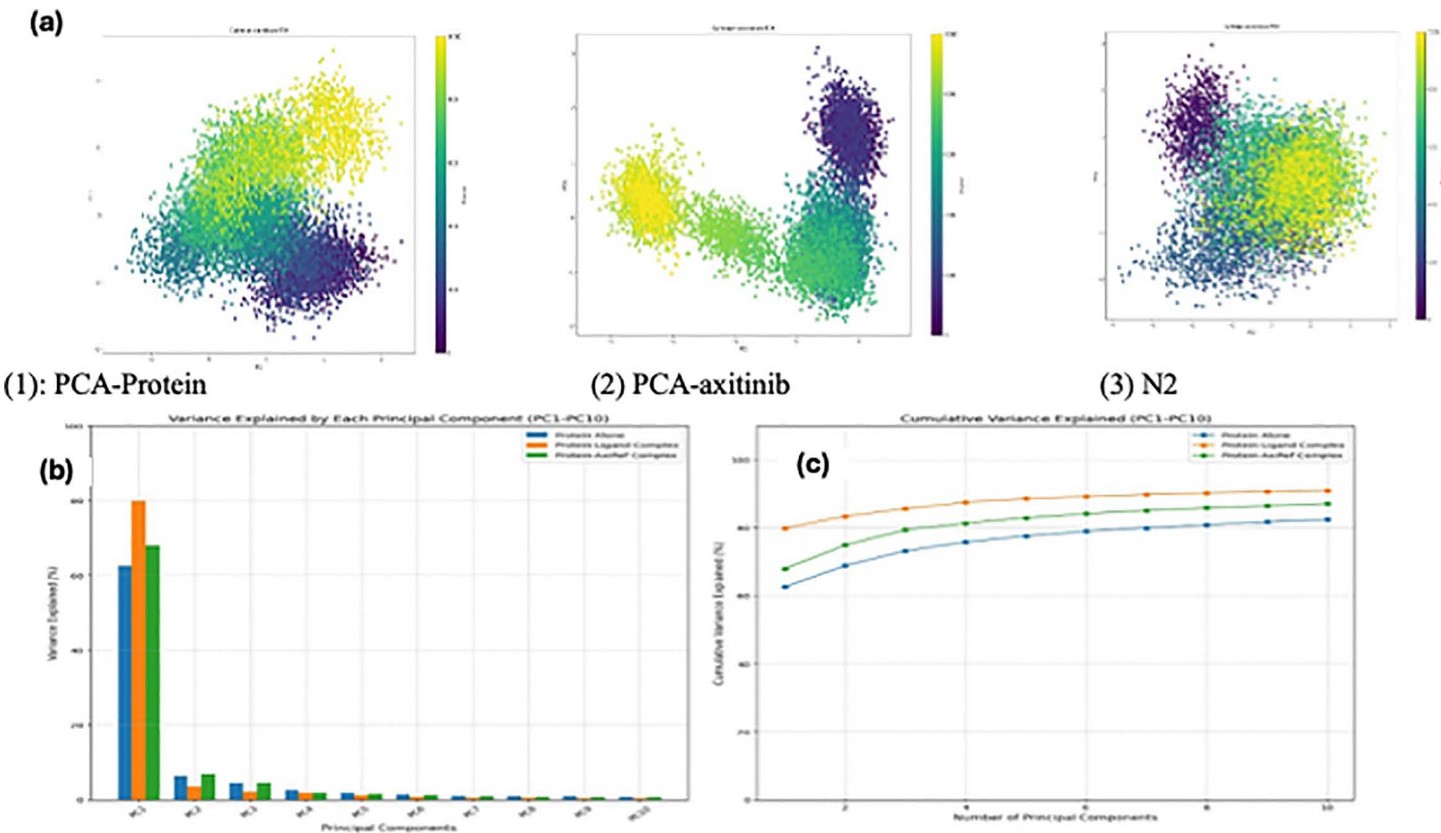

**Fig 12.** (a) principal component analysis of three states ((1) protein only(5EW3)) with (2) Axitinib, and (3) N2. (b) comparison of PCA variance of unliganded protein(5EW3) and liganded (protein-ligand (N2), protein-Axitinib).

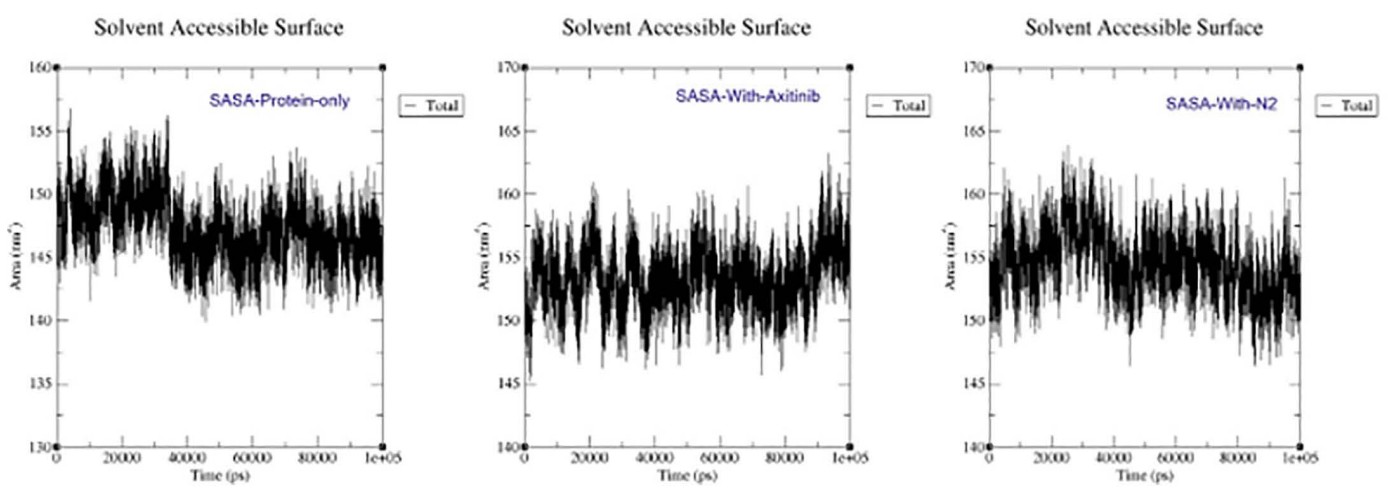

**Fig 13.** SASA Comparison: Protein Only (5EW3), Axitinib, and N2.

during binding. Notably, the N2 shows slightly greater binding consistency than Axitinib and less fluctuation, suggesting a stronger or more favorable interaction profile (Fig 14).

### Radius of Gyration (Rg)

The Rg values indicate the overall compactness and structural expansion of the protein. In the presence of ligands, particularly the new ligand N2, the Rg values increased compared to the apo (ligand-free) protein. This suggests that ligand binding induces a conformational expansion of the protein structure. Such an increase in Rg reflects a structural rearrangement that may influence the protein's functional dynamics. Therefore, the ligand's presence, especially N2, significantly impacts the protein's structural behavior and potentially biological activity (Fig 15).

### Root Mean Square Fluctuation (RMSF) analysis

RMSF is a measure used in molecular dynamics to assess the flexibility of individual residues in a protein. The simulation evaluated the flexibility of residues in the free protein, protein-Axitinib complex, and protein-N2 complex. The RMSF profile of the free protein and Axitinib-bound complex showed significant fluctuations at several regions, particularly around atom ~2000, indicating higher flexibility. In contrast, the N2-bound complex exhibited reduced fluctuations across most regions, suggesting enhanced structural stability upon binding of the N2 compound. This indicates that the N2 binding potentially stabilizes the protein structure more effectively than Axitinib (Fig 16).

### H-bond analysis

Hydrogen bonds are significant for stabilizing ligands in the receptor's pocket. Fig 15 shows the H-bonds between the protein and axitinib (left) and N2 (right) amid the recreation. Axitinib ranges from 0 to 8 bonds (average 4), with relative solidness (4–5 bonds) and minor changes, demonstrating a steady interaction. N2 ranges from 0 to 7 (average 2–4) (Fig 17).

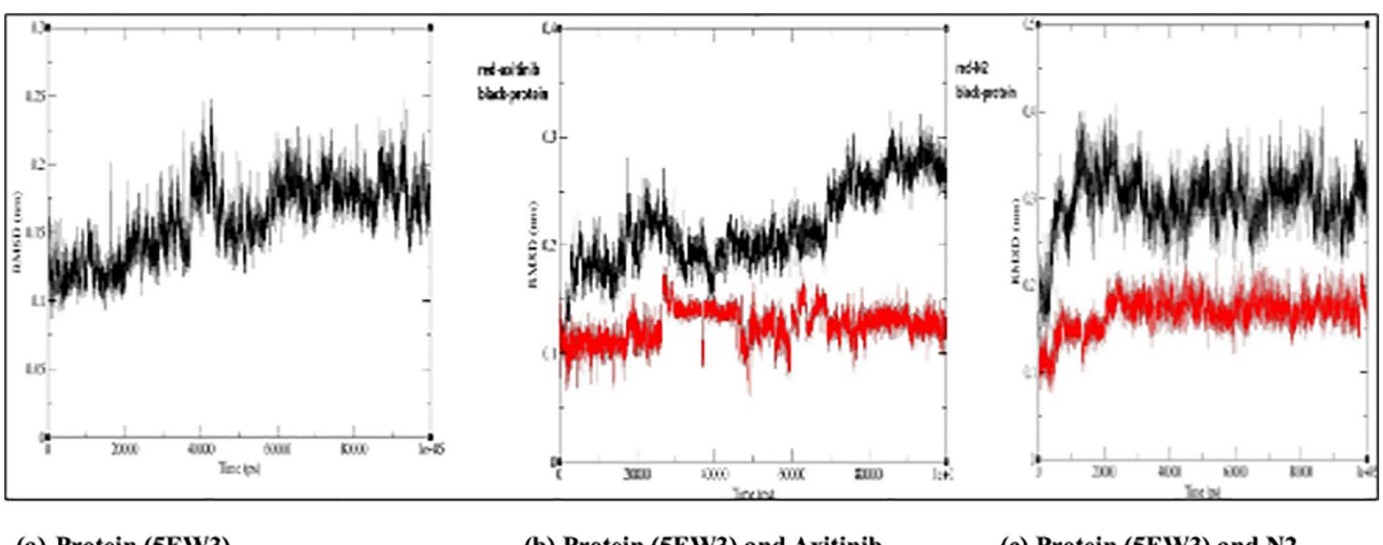

(a) Protein (5EW3)   (b) Protein (5EW3) and Axitinib   (c) Protein (5EW3) and N2

**Fig 14.  RMSD Comparison: Protein Only (5EW3), Axitinib and N2.**

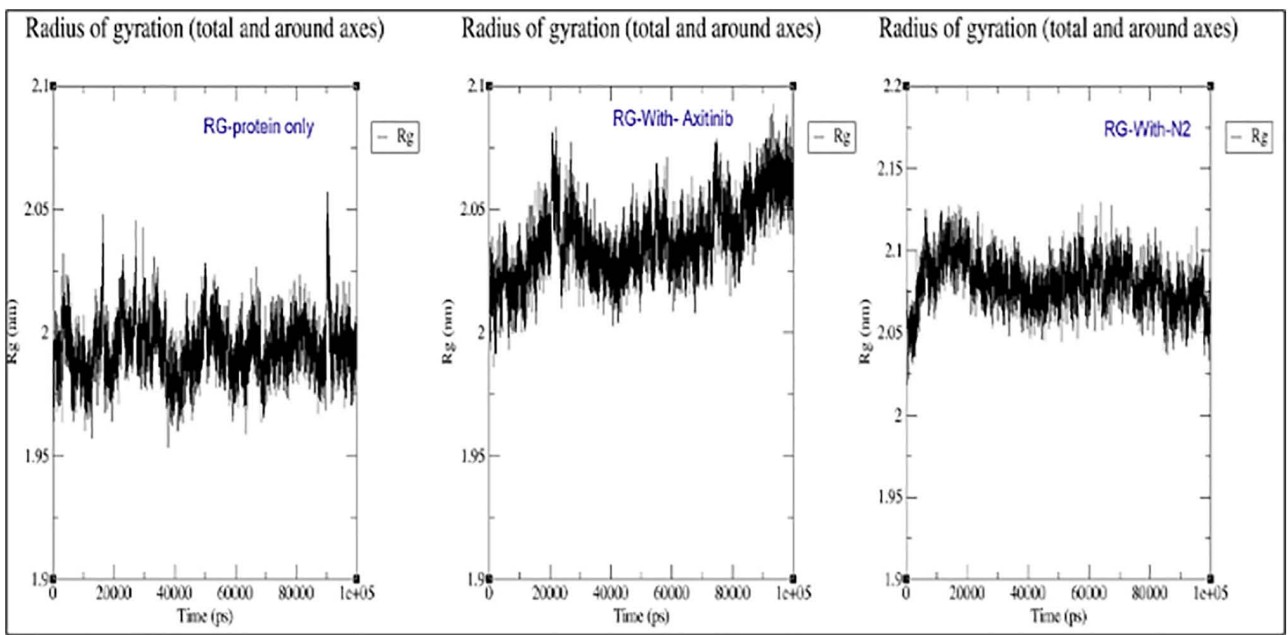

**Fig 15. Rg Comparison: Protein Only(5EW3), Axitinib and N2.**

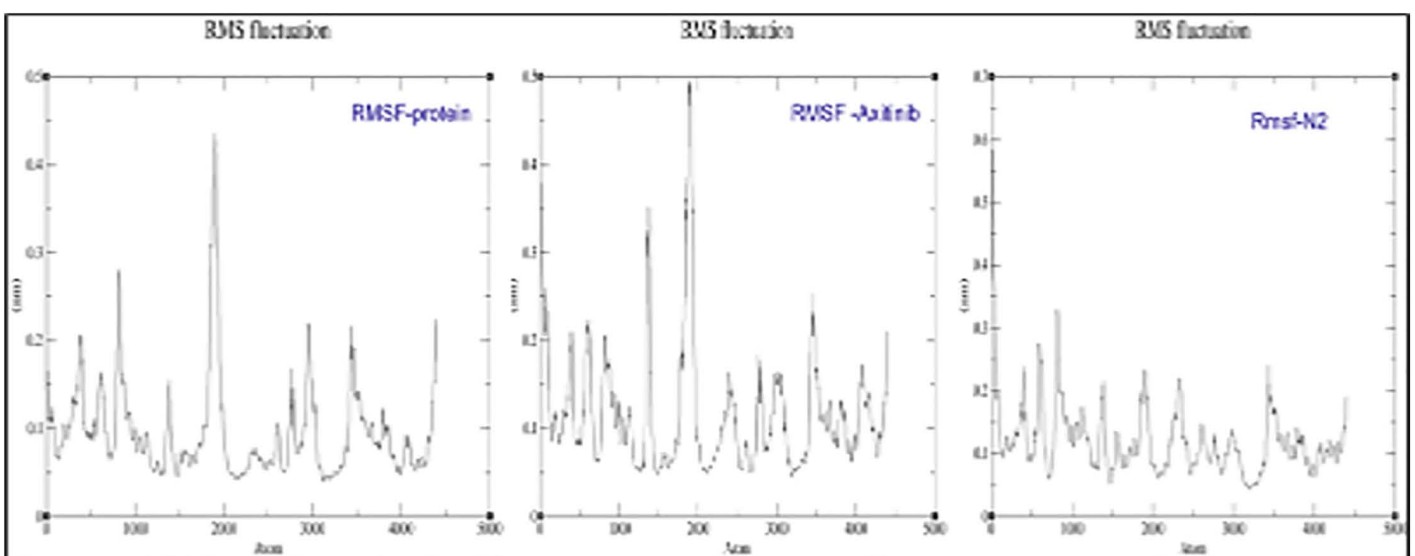

**Fig 16. Root Mean Square Fluctuation (RMSF) Analysis Reveals Enhanced Structural Stability of the Protein(5EW3) upon Binding with N2 Compared to Axitinib.**

## Discussion

Angiogenesis, an essential physiological process for generating new blood vessels, is crucial in embryonic development, wound healing, and ovulation pathways [43]. The direct approach involves regulating the capacity of vascular endothelial

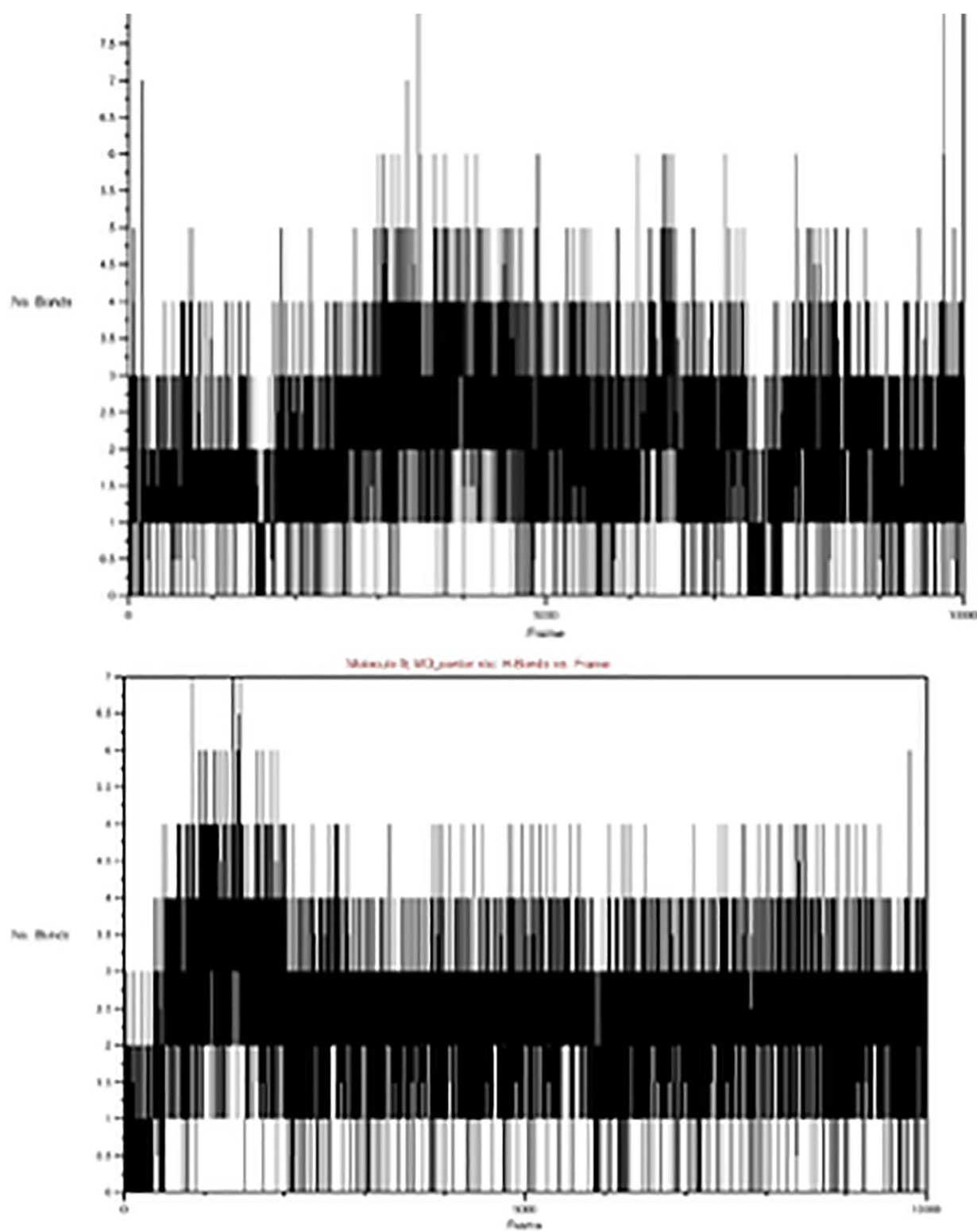

**Fig 17. Hydrogen Bonds Between the Protein(5EW3) and Axitinib, N2.**

cells to proliferate, move, and respond to angiogenic stimuli such as VEGF. The indirect pathway influences the expression and activity of angiogenic factors that promote angiogenesis. This includes regulating receptor expression on endothelial cells, such as chemokine receptor 7 (CCR-7) and the tyrosine kinase receptor IGF-IR [43].

This study examined the anti-angiogenic potential using the RAR test. The N2 demonstrated significant inhibitory action at 200 µg/mL, yielding over 95% mean inhibition. The compound showed notable inhibitory efficacy at 6.25 µg/mL, with $14 \pm 2.51\%$ suppression of vessel development. Bevacizumab demonstrated significant inhibitory action at 100 µg/mL, yielding $95.8 \pm 1.3$ mean inhibition of vessel growth. A dose-response relationship refers to the change in the intensity of a drug's effect as the dose of the drug varies. In this specific case, the relationship for the N2 is determined by assessing how different concentrations influence the contraction or relaxation of vascular tissues, such as rat aorta rings. A dose-response curve has been established to determine the concentration that blocks 50% of blood vessels [2].

Employing rat aorta rings in ex vivo tests enable researchers to investigate the angiogenic process within a controlled setting that closely resembles in vivo conditions. Achieving an $IC_{50}$ value in this model enhances the credibility of results concerning a compound's efficacy in blocking angiogenesis in biological systems. The $IC_{50}$ in this study was 18.91 µg/mL. An $IC_{50}$ value below 20 µg/mL signifies that the compound has considerable potency. Compounds exhibiting low $IC_{50}$ values are frequently investigated for therapeutic purposes, particularly in situations where abnormal angiogenesis exacerbates disease progression, such as cancer, diabetic retinopathy, and other angiogenesis-dependent disorders. Consequently, a molecule with an $IC_{50}$ below 20 µg/mL may signify a potential candidate for advanced research as an anti-angiogenic drug [44].

The CAM assay is a classical technique for the in vivo study of angiogenesis [24]. The present study revealed that CAM treated with 10 µL of the N2 and 10 µL of Bevacizumab exhibited a notable anti-angiogenic effect. Numerous blood vessels treated with the N2 and Bevacizumab ceased to emanate from beneath the disc containing 10 µL of each compound. The blood vessels treated with each compound exhibited a sparsity, disarray, and a pale-yellow hue. These findings indicate that the N2 and Bevacizumab exhibited a significant anti-angiogenic effect *in vivo*. In the 1970s, Folkman suggested targeting blood vessels within tumors to deprive them of nutrients and oxygen [45]. Nevertheless, tumor vasculature's leaky and disordered architecture leads to hypoxic zones' re-emergence as tumors expand [46]. Consequently, recurrent hypoxia leads to the overexpression of HIF-1α, resulting in heightened activation of HIF target genes associated with angiogenesis, metabolism, radioresistance, metastasis, and chemoresistance [47]. Moreover, targeting angiogenesis with VEGF monoclonal antibodies enhances the hypoxic area, leading to HIF-1α accumulation and subsequent upregulation of HIF target genes, including VEGF [48]. Therefore, the cumulative effect of reducing HIF-1α presents greater promise for suppressing tumor growth than addressing angiogenesis in isolation. 1,2,3-triazole derivatives exhibit significant inhibitory effects on HIF-1α transactivation and stability, indicating anti-angiogenic and anti-tumor properties. Moreover, incorporating gefitinib increased the efficacy of 1,2,3-triazole derivatives in anti-tumor activity by reducing hypoxia-induced tumor proliferation [49]. The research demonstrated that the N2 suppressed VEGF-induced proliferation and tube formation in HUVECs. The quantitative synthesis of VEGF signaling proteins and the expression of membrane-bound VEGFR-2 tyrosine kinase were examined in treated human umbilical vein endothelial cells (HUVECs) to clarify the mechanisms responsible for the anti-angiogenic effects of the N2 [20]. The findings indicated that the N2 suppressed VEGF-induced proliferation of HUVECs in a concentration-dependent manner. The data showed that the N2 reduced VEGF-induced endothelial cell growth at an IC50 of 68.85 µg/mL. The VEGF signaling pathway in endothelial cells is crucial for both *in vivo* and *in vitro* angiogenesis. VEGF stimulates the proliferation, migration, and development of capillary-like structures in endothelial cells by producing and activating VEGFR-2 tyrosine kinase [50].

Antiproliferative testing *in vitro* is the initial phase in evaluating prospective anti-cancer agents. This assay employs a cell line that offers benefits, including reduced test material requirements and a shorter duration. The anti-cancer activity is indicated by the percentage of inhibition and the $IC_{50}$ value. A greater proportion of inhibition signifies a more potent

inhibitory function [51]. The results showed that the percentage of inhibition rises with increasing concentrations of the N2. A disparity exists in inhibitory activity against MCF-7 cells and HCT-116 cell proliferation, respectively. The anti-cancer activity is demonstrated by the $IC_{50}$ value (μg/mL). An $IC_{50}$ value greater than 100 indicates that a chemical has anti-cancer activity. An $IC_{50}$ value between 100 and 300 indicates moderate anti-cancer activity, while an $IC_{50}$ value beyond 300 denotes an inert substance [51,52].

The results indicated IC50 values of 124.75 and 112.31 μg/mL for the N2 on MCF-7 and HCT-116 cells, respectively. This finding suggests that the N2 may be developed into an anti-cancer agent.

Consequently, the sensitivity of HCT-116 and MCF-7 cells to this chemical reflects their distinctive genetic characteristics. Further testing is necessary to gain a deeper understanding of the cellular processes affected by the chemical. This investigation demonstrated that the substance effectively inhibited two critical characteristics required for cancer cell growth and proliferation: the capacity for colony formation and the ability to migrate for metastasis. The extract may inhibit these essential characteristics of HCT-116 and MCF-7 cells. Metastasis in cancer cells is a multifaceted biological process characterized by the invasion and migration of these cells, which serves as a primary cause of mortality in cancer patients. Multiple organizations are currently dedicated to developing suitable pharmaceutical options to inhibit metastasis and control cancer growth [53]. The results presented herein show that N2 exhibits anti-breast and anti-colon cancer properties, warranting further investigation for a deeper understanding and potential therapeutic development.

The pharmacological importance of triazole and quinoline frameworks influenced the design of the N2. Triazole derivatives are extensively studied for their ability to stabilize interactions with target enzymes through hydrogen bonding, rendering them valuable pharmacophores in the development of anti-cancer drugs. Quinoline moieties are recognized for their anti-cancer properties, particularly in their ability to interfere with cell cycle regulators and DNA replication mechanisms. The combination of these two pharmacophoric cores was postulated to produce a drug with enhanced anti-cancer efficacy targeting pivotal molecular pathways in HCT116 and MCF7 cancer cells.

Measuring gene expression levels has become fundamental in most molecular biology laboratories. Measuring cellular RNA quantifies the extent of gene expression. Real-time RT-PCR is an effective method for quantifying gene expression. The quantitative metric for real-time PCR is the threshold cycle (CT). The CT refers to the PCR cycle at which the fluorescence signal of the reporter dye exceeds a specified threshold. Displaying data as the CT ensures that the PCR is in the exponential amplification phase. The CT value exhibits an inverse correlation with the amplicon quantity in the reaction; hence, a lower CT signifies a higher amplicon concentration. Real-time PCR data is expressed proportionately to another gene in relative quantification, typically called an internal control [34].

The investigation of *VEGFR-2* gene expression in this work indicated that Kaposi's sarcoma (KS) cells treated with the N2 exhibited a decrease in *VEGFR-2* gene expression compared to the control cells at doses of 50 and 100 μg/mL, but this reduction was not statistically significant. Conversely, significant heterogeneity was noted at 200 μg/mL concentrations concerning their effect on *VEGFR-2* gene expression. The N2, at the final two doses, demonstrated efficacy in suppressing gene expression. It impeded the proliferation, migration, and tube formation of KS cells by obstructing the binding of VEGFR-2 to the surface receptors on KS cells and reducing the expression of VEGFR-2 receptors. Notably, the positive control, bevacizumab, exhibited a significantly greater reduction in *VEGFR-2* gene expression than the N2, even at its highest tested dose (200 μg/mL).

The triazolo-quinoline scaffold in the investigated N2 mirrors established pharmacophoric features of VEGFR-2 inhibitors, including hinge-binding via the quinoline ring and potential aspartate-phenylalanine-glycine (DFG) motif interactions via the triazole moiety. Structurally, this is comparable to known inhibitors like axitinib and sorafenib, supporting its rational design as a VEGFR-2-targeted anticancer agent [4]. The N2 investigated in this study shares critical pharmacophoric elements with reported VEGFR-2 inhibitors, such as the hinge-binding quinoline ring and hydrophobic and H-bonding moieties analogous to those in quinoxaline derivatives [5]. These structural similarities support the rational design of N2 as a VEGFR-2 inhibitor and are consistent with previously validated scaffolds showing promising anticancer and VEGFR-2 inhibitory activity.

VEGF is a principal agonist of angiogenesis. It is a secreted protein that promotes endothelial cell proliferation and vascular permeability. VEGFR-2 is an essential regulator of angiogenesis in both physiological and pathological conditions. VEGF attaches to receptors, triggering signaling pathways that lead to proliferation, migration, survival, and increased vascular permeability. VEGF and its receptors were the most often targeted molecules by anti-angiogenic treatments across various tumor growth progressions [54]. Targeting angiogenesis for illness treatment was proposed more than 50 years ago. Bevacizumab, the inaugural medication designed to inhibit VEGF selectively, was created for the treatment of cancer and neovascular ocular disorders. Investigating and finding novel chemicals that may demonstrate a positive effect as anti-angiogenic agents is a crucial field of research [55]. This study examined the characteristics of a newly identified chemical, the N2, and evaluated its efficacy in suppressing angiogenesis.

We utilized the SwissADME and pkCSM pharmacokinetics servers to compute the ADME/Tox-related descriptors for the N2. These web servers were chosen due to their accessibility and their provision of comprehensive computational techniques for assessing the pharmacokinetics and toxicity of small compounds globally. SwissADME includes methods selected for their robustness, efficiency, and clarity of interpretation. The pkCSM pharmacokinetics server facilitates rapid and dependable predictions of ADME/Tox characteristics, constructed by meticulously choosing data sets and documented methodologies in the literature [35].

Six physicochemical parameters are considered: lipophilicity, size, polarity, solubility, flexibility, and saturation. A physicochemical range for each axis was established using descriptors derived from references [56] and [57], depicted as a pink region within which the molecule's radar plot must fit to be considered drug-like. Positioning the cursor over the radar yields further information on the descriptors. The SwissADME and pkCSM pharmacokinetics services compile fundamental molecular and physicochemical attributes, including molecular weight (MW), molecular refractivity (MR), specific atom type counts, and polar surface area (PSA). The PSA is determined using the fragment-based method known as topological polar surface area (TPSA), which accounts for the OH group. This has demonstrated utility as a descriptor in numerous models and rules for rapidly estimating certain ADME features, particularly for traversing biological barriers like absorption and brain penetration [36].

The partition coefficient between n-octanol and water ($log\ P_{o/w}$) serves as the traditional indicator of lipophilicity. It possesses a distinct area in SwissADME due to the paramount significance of this physicochemical feature for pharmacokinetic drug development [58]. Various computer approaches for estimating log Po/w have been developed, exhibiting differing performance across distinct chemical datasets. It is customary to employ numerous predictors to either identify the best precise approaches for a specific chemical series or to produce consensus estimations. The models underlying the predictors must exhibit maximal diversity to enhance prediction accuracy via consensus $log\ P_{o/w}$. The consensus $log\ P_{o/w}$ is the arithmetic average of the values forecasted by the five suggested approaches.

Specialized models, detailed in the pharmacokinetics section, assess the distinct ADME characteristics of the studied chemical. A multiple linear regression model is designed to estimate the skin permeability coefficient (Kp). It is derived from Potts and Guy [59], and it was discovered that Kp has a linear correlation with molecule size and lipophilicity ($R2 = 0.67$). A lower log Kp value (measured in cm/s) indicates less skin permeability of the molecule. The log Kp of the N2 was −6.69, indicating reduced skin penetration. Descriptors were predicted using pkCSM-pharmacokinetics. Another important consideration is P-glycoprotein, an ATP-dependent efflux pump in various human tissues. All freshly synthesized chemicals were expected to be substrates of P-glycoprotein.

The blood-brain barrier (BBB) is a complex structure that separates the central nervous system (CNS) from peripheral tissues. It regulates the exchange of materials, nutrients, and cells between the blood and the brain to maintain homeostasis within the central nervous system (CNS). Additionally, the BBB plays a role in eliminating cellular metabolites and toxins from the brain into circulation [60]. Understanding the interaction between chemicals and cytochromes P450 (CYP) is crucial. This isoenzyme superfamily is essential in medication clearance via metabolic biotransformation [61]. It has been suggested that CYP and P-glycoprotein (P-gp) can work synergistically to process small molecules, thus enhancing

the protection of tissues and organisms. It is estimated that 50–90% of therapeutic molecules are substrates of five major isoforms (CYP3A4, CYP1A2, CYP2C9, CYP2D6, and CYP2C19) [62].

The inhibition of these isoenzymes significantly contributes to pharmacokinetics-related drug-drug interactions, which can lead to toxicity or other undesirable side effects due to reduced drug clearance and the accumulation of the drug or its metabolites [63]. Various inhibitors for the CYP isoforms have been identified. Some drugs affect multiple CYP isoforms, while others exhibit selectivity for particular isoenzymes. Consequently, it is essential in drug development to predict whether a chemical may induce significant drug interactions through CYP inhibition and to identify the specific isoforms involved.

Drug-likeness is determined through structural or physicochemical evaluations of compounds sufficiently advanced in development to be considered oral drug candidates. Lipinski's rule of five suggests that optimal absorption or permeation is likely when the molecular weight (MW) is less than 500 Da, the number of hydrogen bond donors (HBDs) is fewer than 5, the Log P is below 5, and the number of hydrogen bond acceptors (HBAs) is less than 10, as supported by molecular docking studies [64]. The Swiss Target Prediction is a web service that employs approaches for assessing chemical similarity using molecular fingerprints, also referred to as 2D similarity. Compounds that closely resemble one another based on these measures generally have a heightened tendency to engage with analogous molecular targets. Given the intricate nature of molecular recognition, which encompasses ligand conformation and electrostatics, three-dimensional structural similarity approaches have been developed to assess the spatial comparison of molecules. Current results demonstrate that the amalgamation of 2D and 3D similarity assessments significantly enhances the accuracy of target prediction, especially for novel compounds that do not belong to well-studied chemical classes [65]. While known active components can provide confidence in the prediction, laboratory validation remains necessary to confirm the compound's efficacy against the target protein.

Molecular docking simulations are computational procedures used to test the binding of a molecule to its target. These studies are widely used in the literature to evaluate the *in silico* activities of a designed molecule [38]. VEGF2 receptors undergo autophosphorylation at the tyrosine kinase domain, following VEGF binding, triggering downstream processes that promote vascular proliferation and migration. These processes are crucial in wound healing but can contribute to abnormal angiogenesis and vasculogenesis in various clinical situations, including rheumatoid arthritis, psoriasis, ocular issues, and tumor proliferation [66]. Axitinib, a VEGFR-specific angiogenesis inhibitor approved by the FDA, is known for its minimal adverse reactions and enhanced safety profile. Therefore, our study used axitinib as a reference drug to discover potential VEGFR-2 inhibitors [67]. Table 7 shows that the 2D interaction maps of the ligands interact hydrophobically and establish hydrogen bonds with important residues, such as GLU885 and LYS868 (numbering may vary). But axitinib has a lower binding energy because it produces more complex interactions, such as π-π stacking and hydrophobic contacts. The 3D interaction maps corroborate these results, revealing that axitinib is more entrenched and stable within the binding site than the N2, which forms meaningful connections but has a smaller network and fewer interaction points. Based on these findings, axitinib appears to have a better binding profile when docked, even though the N2 shows significant binding.

Molecular dynamics simulation is an effective computational strategy that recreates the time-dependent behavior of atomic frameworks by joining Newton's conditions of movement, regularly utilizing constrain areas like Assisted Model Building with Energy Refinement (AMBER) or Chemistry at Harvard Macromolecular Mechanics (CHARMM) [68]. It gives atomic-level bits of knowledge into conformational changes and interactions.

The interactions between the 5EW3 receptor, axitinib, and compound N2 are shown in Fig 11 in both the first and last frames. Consistent hydrogen bonding with GLU65 and LYS48, as well as hydrophobic interactions with CYS169, all contributed to the structural stability of the axitinib-5EW3 complex. Hydrophobic interactions involving CYS169 and LEU159 were also seen in the N2-5EW3 complex, in addition to important hydrogen bonds with GLU65 and LYS48. These persistent contacts support the moderate yet stable RMSD and radius of gyration (Rg) values observed during simulation. Vital Component Investigation (PCA) diagonalizes the covariance matrix of atomic displacements, extracting dominant collective motions (e.g., breathing or twisting) from trajectories [69]. According to PCA, each of the three systems

exhibited unique patterns of motion (Fig 12). With PC1 being responsible for 62.6% of the motion, the apoprotein demonstrated a significant level of conformational variation, which is indicative of its intrinsic flexibility. N2 complex dynamics were drastically limited, with PC1 resolving 79.8% of the total motion, indicating a protein shape that was probably rigidified due to strong ligand binding. The axitinib complex exhibited a balanced dynamic profile with modest structural flexibility, while PC1 contributed 68.0%. As illustrated in Fig 13, this increase suggests a minor conformational expansion upon ligand binding. The N2-bound protein exhibited a comparable SASA pattern to that of Axitinib, indicating similar solvent interaction and structural relaxation characteristics, with only subtle differences observed. Root Mean Square Deviation (RMSD) tracks basic deviations from a reference (e.g., starting or precious stone structure), demonstrating solidness or merging over time [70]. As illustrated in Fig 14, all systems reached equilibrium following the initial simulation phase, with the protein undergoing only minor conformational changes upon ligand binding. Axitinib and the N2 showed stable protein-ligand interactions with low RMSD fluctuations. Notably, compared to axitinib, the N2 exhibited a slightly lower variation, which may indicate a stronger structural stability of the protein and more consistent binding. These results support the hypothesis that both ligands bind stably to the target protein, with the N2 demonstrating slightly enhanced dynamic retention within the active site. The Radius of Gyration (Rg) measures a molecule's compactness by calculating the root-mean-square distance of atoms from the center of mass, reflecting folding/unfolding dynamics [71]. The Rg profiles of the protein are shown in Fig 15, both in its unbound and complexed states with axitinib and the N2. The Rg values were reasonably constant in the ligand-free protein, suggesting a compact conformation. Upon ligand binding, particularly with the N2, the Rg values increased slightly, reflecting a modest conformational expansion. This provides more evidence that structural rearrangement is induced by ligand contact, with the N2 showing a more pronounced effect. This could have implications for the flexibility and function of proteins. Root Mean Square Fluctuation (RMSF) captures residue-level flexibility by averaging positional fluctuations, highlighting mobile regions like loops [72]. As shown in Fig 16, the free protein and the axitinib-bound complex displayed notable residue fluctuations, particularly around atom ~2000, indicating regions of increased flexibility. On the other hand, the complex that was attached to the N2 showed fewer fluctuations across the most of residues, which could indicate that the structure was more rigid and stabilized after binding the ligand. These findings imply that N2 may confer more pronounced stabilizing effects on the protein structure compared to axitinib. Hydrogen bonds (H-bonds), critical for structural integrity, are analyzed via geometric criteria (distance $< 3.5$ Å, angle $> 120°$) and their dynamics [73]. Fig 17 illustrates the hydrogen bonding profiles of axitinib and N2 with the target protein throughout the simulation. Axitinib consistently formed 4–5 hydrogen bonds on average, with minimal fluctuation, indicating a stable interaction. In contrast, the N2 formed 2–4 hydrogen bonds on average, with slightly broader variation. While both ligands maintained stable binding, axitinib exhibited a marginally stronger hydrogen bonding profile.

## Conclusions

This study concludes that N2 is an antiproliferative drug. The drug exhibited significant anti-angiogenic action in the ex vivo rat aorta and in vivo CAM assays. It inhibited VEGF-induced proliferation and tube formation in HUVECs and showed anti-proliferative effects against HCT116 and MCF7 cancer cell lines by inhibiting the *VEGFR-2* gene. Molecular docking analyses demonstrated robust binding affinities with the target protein, corroborated by advantageous interactions with critical amino acid residues. Molecular dynamics simulation findings indicate that axitinib and N2 stabilize VEGFR-2 structure upon binding, with the N2 inducing greater conformational rigidity and reduced flexibility, potentially enhancing its inhibitory efficacy. The ADME investigation revealed favorable pharmacokinetic characteristics, indicating the compound's potential for therapeutic development.

The findings of this study establish a basis for further investigation of the phenyl isoserine derivative as an anti-cancer drug. Subsequent research may focus on structural alterations to improve its potency and selectivity, alongside preclinical trials to assess its therapeutic efficacy and safety in vivo. Incorporating these findings into the broader framework of cancer therapy development may provide novel pathways for viable treatment options.

## Acknowledgments

The authors thank the College of Medicine, Al-Nahrain University, for their essential support in enabling this research. Establishing a fully equipped laboratory and promoting scientific inquiry significantly facilitated the effective completion of this investigation. Their commitment to promoting research excellence is greatly valued.

## Informed consent statement

Not applicable.

## Author contributions

**Conceptualization:** Nawar Raad Hussein.

**Data curation:** Nawar Raad Hussein.

**Formal analysis:** Nawar Raad Hussein.

**Funding acquisition:** Nawar Raad Hussein.

**Investigation:** Nawar Raad Hussein.

**Methodology:** Nawar Raad Hussein, Hayder B. Sahib, Zahraa Sabbar Omran.

**Project administration:** Nawar Raad Hussein.

**Supervision:** Hayder B. Sahib, Zahraa Sabbar Omran.

**Validation:** Nawar Raad Hussein.

**Visualization:** Hayder B. Sahib, Zahraa Sabbar Omran.

**Writing – original draft:** Nawar Raad Hussein.

**Writing – review & editing:** Nawar Raad Hussein.

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
