## [Decision Letter · Decision Letter 0]

28 Mar 2025

PONE-D-25-02987The Anti-Angiogenic and Anti-Proliferative Activity of an N-Benzoyl-3-phenyl Isoserine Derivative Containing 1,2,3-Triazole and Quinoline RingsPLOS ONE

Dear Dr. Hussein,

Thank you for submitting your manuscript to PLOS ONE. After careful consideration, we feel that it has merit but does not fully meet PLOS ONE’s publication criteria as it currently stands. Therefore, we invite you to submit a revised version of the manuscript that addresses the points raised during the review process.

If applicable, we recommend that you deposit your laboratory protocols in protocols.io to enhance the reproducibility of your results. Protocols.io assigns your protocol its own identifier (DOI) so that it can be cited independently in the future. For instructions see: https://journals.plos.org/plosone/s/submission-guidelines#loc-laboratory-protocols. Additionally, PLOS ONE offers an option for publishing peer-reviewed Lab Protocol articles, which describe protocols hosted on protocols.io. Read more information on sharing protocols at . Additionally, PLOS ONE offers an option for publishing peer-reviewed Lab Protocol articles, which describe protocols hosted on protocols.io. Read more information on sharing protocols at https://plos.org/protocols?utm_medium=editorial-email&utm_source=authorletters&utm_campaign=protocols..

We look forward to receiving your revised manuscript.

Kind regards,

Hazem Elkady

Academic Editor

PLOS ONE

Journal Requirements:

3. To comply with PLOS ONE submissions requirements, in your Methods section, please provide additional information regarding the experiments involving animals and ensure you have included details on (1) methods of sacrifice, (2) methods of anesthesia and/or analgesia, and (3) efforts to alleviate suffering.

Reviewers' comments:

Reviewer's Responses to Questions

**Comments to the Author**

1. Is the manuscript technically sound, and do the data support the conclusions?

Reviewer #1: Partly

2. Has the statistical analysis been performed appropriately and rigorously? 

Reviewer #1: I Don't Know

3. Have the authors made all data underlying the findings in their manuscript fully available?

Reviewer #1: Yes

4. Is the manuscript presented in an intelligible fashion and written in standard English?

Reviewer #1: Yes

5. Review Comments to the Author

Reviewer #1: The efforts exerted in this current work are so appreciated. However, some points need to be addressed. So, a major revision may be required to improve the manuscript:

1. The abstract should be systematically improved. It is recommended to give a brief background on cancer and the importance of selected target for cancer treatment, then the importance and aim of your work, then the experiments conducted and their outstanding results, then finally write what you conclude from this work.

2. Many typos and grammatical errors need fixation. The whole manuscript is recommended to to be revised by a native English speaker.

3. Some abbreviations need to be defined for the first time. Please check

4. The introduction is too long. It needs to be shorter and more focusing.

5. The authors should illustrate the essential pharmacophores of VEGFR-2 inhibitors and some previously reported VEGFR-2 inhibitors bear the same scaffold of the derivative under investigation.

6. It is highly recommended to conduct molecular docking using PDB codes that include triazolo quinoline derivatives as co-crystallized ligand to get more reliable results.

7. All biological investigations should be carried out using positive controls.

8. Why authors have selected axitinib, in particular, as a reference drug for molecular docking study.

9. The molecular docking program validation should be carried out.

10. It is highly recommended to conduct molecular dynamic simulations for N2 compound at VEGFR-2 receptors in comparison to a reference drug.

6. PLOS authors have the option to publish the peer review history of their article (what does this mean?). If published, this will include your full peer review and any attached files.). If published, this will include your full peer review and any attached files.

.

Reviewer #1: No

While revising your submission, please upload your figure files to the Preflight Analysis and Conversion Engine (PACE) digital diagnostic tool, https://pacev2.apexcovantage.com/. PACE helps ensure that figures meet PLOS requirements. To use PACE, you must first register as a user. Registration is free. Then, login and navigate to the UPLOAD tab, where you will find detailed instructions on how to use the tool. If you encounter any issues or have any questions when using PACE, please email PLOS at . PACE helps ensure that figures meet PLOS requirements. To use PACE, you must first register as a user. Registration is free. Then, login and navigate to the UPLOAD tab, where you will find detailed instructions on how to use the tool. If you encounter any issues or have any questions when using PACE, please email PLOS at figures@plos.org. Please note that Supporting Information files do not need this step.. Please note that Supporting Information files do not need this step.

---

## [Author Response · Author response to Decision Letter 1]

11 Sep 2025

Response to Reviewers

We sincerely thank the reviewer and the editorial team for their thoughtful and constructive comments on our manuscript. We truly appreciate the time and effort invested in evaluating our work. The feedback has been invaluable in enhancing the clarity, scientific rigor, and overall quality of the manuscript.

All journal requirements have been fully addressed, including formatting per PLoS ONE style, ORCID ID validation, detailed animal ethics methods, abstract consistency, and Supporting Information captions and citations.

In this revised version, we have carefully addressed each of the reviewer’s points in a systematic manner. Below, we provide detailed responses to each comment, highlighting the revisions and rationale behind them. We hope the improvements meet your expectations and contribute positively to the scientific value of our work.

1. The abstract should be systematically improved. It is recommended to give a brief background on cancer and the importance of selected target for cancer treatment, then the importance and aim of your work, then the experiments conducted and their outstanding results, then finally write what you conclude from this work.

We sincerely thank the reviewer for this constructive and insightful suggestion. We greatly appreciate your guidance in helping us improve the structure and clarity of the abstract. In response, we have carefully revised the abstract to follow the recommended structure: starting with a brief background on cancer and the rationale for targeting VEGFR-2, followed by the significance and aim of our study, a concise overview of the conducted experiments and key findings, and concluding with the implications of our results.

The revised abstract:

One of the biggest causes of death globally is cancer, which develops tumors characterized by angiogenesis and excessive cell proliferation. Cancer treatments targeting these pathways, especially those involving vascular endothelial growth factor, have had encouraging results. This study examined the anti-angiogenic and anti-proliferative properties of a phenyl isoserine derivative compound. The ex vivo anti-angiogenic activity was examined by rat aorta ring (RAR) and in vivo chorioallantoic membrane (CAM) assays. The in vitro anti-proliferative properties of the compound were analyzed utilizing Human umbilical vein endothelial (HUVEC), colon cancer, and breast cancer cell lines. Quantitative real-time PCR was used to measure gene expression to evaluate the vascular endothelial growth factor receptor 2 (VEGFR-2) gene expression. An evaluation was conducted on the compound's SwissADME profile, molecular docking, and molecular dynamics simulations against the VEGFR-2 receptor. The phenyl isoserine derivative demonstrated significant anti-angiogenic activity in both the RAR and CAM assays, with a half-maximal inhibitory Concentration (IC50) value of 18.91 μg/mL in the RAR assay. The compound also exhibited anti-proliferative effects against HUVEC, colon cancer, and breast cancer cells, with IC50 values of 68.85, 124.75, and 112.31 μg/mL, respectively. The results of the Real-Time PCR test showed that at a concentration of 200 μg/mL, there was a significant decrease in the expression of the VEGFR-2 gene in Kaposi’s Sarcoma (KS) cells. The compound exhibited membrane permeability and dispersion, as expected based on the SwissADME profile, with a consensus log P o/w value of 3.14, while also indicating potential cytochrome P450 enzyme inhibition and poor water solubility (Log Sw value: -8.34). Molecular docking studies revealed a higher PLP fitness score (91.496) for the compound compared to axitinib (78.748), suggesting a favorable binding to VEGFR-2. However, axitinib demonstrated a more negative binding energy (−14.548 kcal/mol vs. −7.715 kcal/mol), reflecting stronger energetic stabilization under docking conditions. Molecular dynamics simulations showed that legend exhibits stable binding and induces greater structural stabilization of the target protein compared to axitinib, suggesting its potential as a more effective VEGFR-2 inhibitor. The phenyl isoserine derivative was found to have a potent inhibitory effect on angiogenesis and VEGFR-2 gene activity, indicating it could be a promising candidate for anti-cancer medicine.

2. Many typos and grammatical errors need fixation. The whole manuscript is recommended to be revised by a native English speaker.

We express our heartfelt gratitude to the reviewer for this insightful observation. The work has been meticulously reviewed to rectify typographical, grammatical, and stylistic mistakes. We have also carefully evaluated the material to enhance clarity and coherence and worked with a native English speaker to ensure the language reaches the required standard. We hope the revised version indicates a considerable improvement in linguistic quality and readability.

3. Some abbreviations need to be defined for the first time. Please check

We appreciate the reviewer’s attention to detail. We have carefully reviewed the entire manuscript to ensure that abbreviations are clearly defined upon their first occurrence. Thank you for highlighting this important aspect.

4. The introduction is too long. It needs to be shorter and more focusing.

Thank you for this valuable suggestion. Following your recommendation, we have substantially revised the introduction by removing redundant background details and streamlining the content to better focus on the significance of VEGFR-2 as a therapeutic target and the rationale behind our compound design. The revised introduction has been condensed from two pages to one, enhancing clarity and precision.

5. The authors should illustrate the essential pharmacophores of VEGFR-2 inhibitors and some previously reported VEGFR-2 inhibitors bear the same scaffold of the derivative under investigation.

We are grateful for this insightful recommendation. In response, we have now clearly illustrated the essential pharmacophoric features of VEGFR-2 inhibitors in both the Introduction and Discussion sections. Furthermore, we have highlighted how our triazolo-quinoline derivative shares key structural motifs with previously reported VEGFR-2 inhibitors, including hinge-binding heterocycles and hydrophobic scaffolds. Supporting references (e.g., McTigue et al., 2009; and the study on quinoxaline-based VEGFR-2 inhibitors) have been incorporated to substantiate this comparison and reinforce the relevance of our scaffold within this target class.

Part of the introduction:

(VEGFR-2 inhibitors often include a consistent pharmacophore model that includes four critical components: a hinge-binding heteroaromatic ring, a hydrophobic central scaffold, interaction with the aspartate-phenylalanine-glycine (DFG) motif, and a solvent-exposed polar tail. These characteristics facilitate high-affinity and selective inhibition via critical interactions with hinge residues (e.g., Cys919), the hydrophobic pocket, and Asp1046 within the DFG motif. [4]. Many FDA-approved inhibitors of the VEGFR-2 enzyme (axitinib, cabozantinib, lenvatinib, sorafenib, and tivozanib) possess four critical pharmacophoric characteristics: a hinge-binding heteroaromatic ring, a central aromatic linker, a pharmacophoric hydrogen bond donor/acceptor moiety interacting with the DFG motif, and a terminal lipophilic group that occupies the allosteric pocket. Previous studies have effectively integrated these characteristics into quinoxaline-based frameworks, showcasing their promise as VEGFR-2-targeted anticancer drugs [5].

Part of the discussion:

(The triazolo-quinoline scaffold in the investigated N2 compound mirrors established pharmacophoric features of VEGFR-2 inhibitors, including hinge-binding via the quinoline ring and potential aspartate-phenylalanine-glycine (DFG) motif interactions via the triazole moiety. Structurally, this is comparable to known inhibitors like axitinib and sorafenib, supporting its rational design as a VEGFR-2-targeted anticancer agent [4]. The N2 compound investigated in this study shares critical pharmacophoric elements with reported VEGFR-2 inhibitors, such as the hinge-binding quinoline ring and hydrophobic and H-bonding moieties analogous to those in quinoxaline derivatives [5]. These structural similarities support the rational design of N2 as a VEGFR-2 inhibitor and are consistent with previously validated scaffolds showing promising anticancer and VEGFR-2 inhibitory activity.)

References:

4. McTigue M, Murray BW, Chen JH, Deng Y-L, Solowiej J, Kania RS. Molecular conformations, interactions, and properties associated with drug efficiency and clinical performance among VEGFR TK inhibitors. Proceedings of the National Academy of Sciences 2012;109:18281–9. https://doi.org/10.1073/pnas.1207759109.

5. Alanazi MM, Elkady H, Alsaif NA, Obaidullah AJ, Alanazi WA, Al-Hossaini AM, et al. Discovery of new quinoxaline-based derivatives as anticancer agents and potent VEGFR-2 inhibitors: Design, synthesis, and in silico study. J Mol Struct 2022;1253:132220. https://doi.org/10.1016/j.molstruc.2021.132220.

6. It is highly recommended to conduct molecular docking using PDB codes that include triazolo quinoline derivatives as co-crystallized ligand to get more reliable results.

We express our heartfelt gratitude to the reviewer for this invaluable recommendation. We comprehensively searched the Protein Data Bank (PDB) for VEGFR-2 crystal structures co-crystallized with triazolo-quinoline derivatives. Regrettably, to our knowledge, no such constructions are presently accessible. Notwithstanding this constraint, we employed the VEGFR-2 structure (PDB: 5EW3), a validated kinase domain frequently utilized in analogous investigations, and conducted molecular docking using GOLD software with the PLP scoring tool to evaluate our drug's binding affinity and conformation. We utilized the Glide software from the Schrödinger suite to evaluate our drug's binding energy. Additionally, to enhance our docking outcomes, we performed molecular dynamics (MD) simulations to assess the stability of the docked complex over time. We trust that these initiatives demonstrate our commitment to providing thorough and reliable binding interaction evaluations, despite the unavailability of the specified co-crystallized ligand.

Results: the Glide software from the Schrödinger suite to evaluate our drug's binding energy

The 2D and 3D interaction maps of the N2 compound and Axitinib with receptor (5EW3) of vascular endothelial growth factor receptor 2 (VEGFR-2)

2D interaction map 3D interaction map Binding energy (kcal/mol)

N2 compound

-14.548

Axitinib

-7.715

7. All biological investigations should be carried out using positive controls.

We sincerely thank the reviewer for this critical observation. Following standard experimental protocols, we employed bevacizumab as a positive control in our biological studies, which included the rat aortic ring assay, CAM assay, and VEGF gene expression study. Bevacizumab was chosen for its clinically established function as an anti-angiogenic drug that targets VEGF. Including this reference ensured our experimental results' reliability and interpretability. We recognize the necessity of explicitly articulating this in the manuscript and have revised the pertinent portions accordingly. We apologize for the delay in our response, which was necessitated by the time required to provide the positive control with the necessary care and precision.

Results of the use of Bevacizumab as a positive control in biological investigations

100 μg/ml Bevacizumab (positive control) on blood vessel formation in rat aorta rings

Anti-angiogenesis activity of the different concentrations of the N2 compound along with Bevacizumab (positive control) and 1% DMSO (negative control) in ex vivo aortic ring model

The degree of the inhibition zone of blood vessel growth in vivo (CAM) assay: Bevacizumab (positive control)

Fold change in gene expression of the VEGF gene after exposure to different concentrations of the N2 compound, determined using the ΔΔCt method. Bevacizumab (positive control)

8. Why authors have selected axitinib, in particular, as a reference drug for molecular docking study.

Axitinib was selected as a reference compound due to its well-established and FDA-approved role as a selective VEGFR-2 inhibitor used clinically in anti-angiogenic cancer therapy. Structurally, axitinib binds with high affinity to the ATP-binding site of VEGFR-2 and serves as a benchmark in molecular docking studies evaluating novel VEGFR-2-targeting agents. Its crystallographic data are readily available (PDB: 5EW3), enabling accurate comparison of binding interactions and scoring against our test compound. We have clarified this rationale in the revised manuscript to reflect the scientific justification behind our choice better.

9. The molecular docking program validation should be carried out.

We sincerely thank the reviewer for raising this important point. In response, we performed rigorous validation of our molecular docking protocol. Initially, we conducted a re-docking of the co-crystallized ligand into the VEGFR-2 active site (PDB ID: 5EW3), and the resulting Root Mean Square Deviation (RMSD) value between the docked pose and the native conformation was below 2.0 Å, indicating good reproducibility and validating the docking accuracy. To strengthen our findings, we also used Glide software (Schrödinger suite) to evaluate our compound's binding energy and docking interactions, providing an additional layer of precision and cross-platform consistency. Moreover, to simulate fundamental physiological dynamics and validate the stability of ligand-receptor interactions, we performed Molecular Dynamics (MD) simulations, which included:

Root Mean Square Deviation (RMSD)

Radius of Gyration (Rg)

Root Mean Square Fluctuation (RMSF)

Solvent-Accessible Surface Area (SASA)

Hydrogen bond analysis

Principal Component Analysis (PCA)

These simulations demonstrated that our compound maintained a stable interaction within the VEGFR-2 binding site throughout the simulation, supporting the reliability of our docking results. All relevant figures and data are provided in the revised manuscript, and the next point.

10. It is highly recommended to conduct

We sincerely thank the reviewer for this valuable suggestion. In response, molecular dynamics (MD) simulations were successfully performed for the VEGFR-2 receptor (PDB ID: 5EW3) in complex with both the N2 compound and the reference drug, as well as for the unbound receptor. The simulations used GROMACS 2023.0 on a Linux 22.4 system, employing the TIP3P water model for solvation.

To comprehensively evaluate the stability and dynamic behavior of the complexes, we calculated key metrics including Root Mean Square Deviation (RMSD), Radius of Gyration (Rg), Root Mean Square Fluctuation (RMSF), Solvent-Accessible Surface Area (SASA), Hydrogen bonding, and Principal Component Analysis (PCA). These analyses confirmed the structural stability and meaningful interaction of the N2 compound with the receptor, comparable to the reference drug. The full results are now included in the revised manuscript.

Results of Molecular Dynamics simulations:

A

B

Interaction Analysis: (a) The first frame and last frame interaction of (5EW3) with Axitinib. (b) The first frame and last frame interaction of (5EW3) with N2

a

b

(a) principal component analysis of three states (protein only(5EW3)) with Axitinib, N2 compound. (b) comparison of PCA variance of unliganded protein(5EW3) and liganded (protein-ligand (N2 compound), protein-Axitinib)

SASA Comparison: Protein Only (5EW3), Axitinib and N2 compound

RMSD Comparison: Protein Only (5EW3), Axitinib and N2 compound

Rg Comparison: Protein Only(5EW3), Axitinib and N2 compound. Root Mean Square Fluctuation (RMSF) Analysis Reveals Enhanced Structural Stability of the Protein(5EW3) upon Binding with N2 Compound Compared to Axitinib

Hydrogen Bonds Between the Protein(5EW3) and Axitinib, N2 compound

We sincerely appreciate the constructive comments and valuable suggestions provided by the reviewer, which have

---

## [Decision Letter · Decision Letter 1]

15 Oct 2025

PONE-D-25-02987R1The Anti-Angiogenic and Anti-Proliferative Activity of an N-Benzoyl-3-phenyl Isoserine Derivative Containing 1,2,3-Triazole and Quinoline RingsPLOS ONE

Dear Dr. Hussein,

Thank you for submitting your manuscript to PLOS ONE. After careful consideration, we feel that it has merit but does not fully meet PLOS ONE’s publication criteria as it currently stands. Therefore, we invite you to submit a revised version of the manuscript that addresses the points raised during the review process.

If applicable, we recommend that you deposit your laboratory protocols in protocols.io to enhance the reproducibility of your results. Protocols.io assigns your protocol its own identifier (DOI) so that it can be cited independently in the future. For instructions see: https://journals.plos.org/plosone/s/submission-guidelines#loc-laboratory-protocols. Additionally, PLOS ONE offers an option for publishing peer-reviewed Lab Protocol articles, which describe protocols hosted on protocols.io. Read more information on sharing protocols at . Additionally, PLOS ONE offers an option for publishing peer-reviewed Lab Protocol articles, which describe protocols hosted on protocols.io. Read more information on sharing protocols at https://plos.org/protocols?utm_medium=editorial-email&utm_source=authorletters&utm_campaign=protocols..

We look forward to receiving your revised manuscript.

Kind regards,

Hazem Elkady

Academic Editor

PLOS ONE

Journal Requirements:

Reviewers' comments:

Reviewer's Responses to Questions

**Comments to the Author**

1. If the authors have adequately addressed your comments raised in a previous round of review and you feel that this manuscript is now acceptable for publication, you may indicate that here to bypass the “Comments to the Author” section, enter your conflict of interest statement in the “Confidential to Editor” section, and submit your "Accept" recommendation.

Reviewer #1: (No Response)

Reviewer #2: All comments have been addressed

2. Is the manuscript technically sound, and do the data support the conclusions?

Reviewer #1: (No Response)

Reviewer #2: Yes

3. Has the statistical analysis been performed appropriately and rigorously? 

Reviewer #1: (No Response)

Reviewer #2: Yes

4. Have the authors made all data underlying the findings in their manuscript fully available?

Reviewer #1: (No Response)

Reviewer #2: Yes

5. Is the manuscript presented in an intelligible fashion and written in standard English?

Reviewer #1: (No Response)

Reviewer #2: Yes

6. Review Comments to the Author

Reviewer #1: The manuscript entitled "The Anti-Angiogenic and Anti-Proliferative 1 Activity of N-Benzoyl-3-phenyl isoserine derivative containing 1,2,3-triazole and quinoline rings" can be accepted in its current form

Reviewer #2: 1] The manuscript uses multiple names: “N2 compound”, “phenyl isoserine derivative”, “N-Benzoyl-3-phenyl isoserine derivative”, etc. Choose one short label (e.g., “N2”) and use it consistently after the first full chemical name. Ensure the first instance includes the full IUPAC name and a 2D structure (Fig. 1).

2] The manuscript alternates between “VEGF gene”, “VEGFR-2 gene”, and “VEGF” in ways that may confuse readers. Use standard nomenclature (e.g., VEGFR-2 for the receptor; VEGF-A for the ligand if measured). Italicize gene symbols where journal style requires it and be consistent throughout.

3] Some abbreviations (e.g., RAR, CAM, HUVEC, MD, RMSD) are used before being defined. Define each abbreviation at first mention and consider adding an abbreviations list.

4] Minor typos and awkward phrases remain (e.g., “legend exhibits stable binding”, probably “ligand”). Fix obvious typos and awkward sentences; ensure subject/verb agreement. A quick pass by a native English editor or use of proofreading software would resolve these.

5] Some panels are low resolution and legends lack detail (e.g., exact n, statistical test used, meaning of error bars). Upload high-resolution figures, label panels (A, B, C), and update each legend to include sample size (n), test used, definition of error bars (SD or SEM), and exact p-value formatting.

6] The manuscript uses “P ≤ 0.05” and “P < 0.05” inconsistently and often gives only threshold statements. Use consistent lowercase or uppercase p per journal style (commonly “p”) and report exact p-values where feasible (e.g., p = 0.0001), and always state which test was used in figure legends.

7] pH was shown as “7.5” and elsewhere as “7.5,” decimal reporting varies. Standardize number formatting (e.g., one or two decimal places consistently) and ensure percent signs have spaces.

7. PLOS authors have the option to publish the peer review history of their article (what does this mean?). If published, this will include your full peer review and any attached files.). If published, this will include your full peer review and any attached files.

.

Reviewer #1: No

Reviewer #2: **Yes:** Mahmoud S. ElkotamyMahmoud S. Elkotamy

While revising your submission, please upload your figure files to the Preflight Analysis and Conversion Engine (PACE) digital diagnostic tool, https://pacev2.apexcovantage.com/. PACE helps ensure that figures meet PLOS requirements. To use PACE, you must first register as a user. Registration is free. Then, login and navigate to the UPLOAD tab, where you will find detailed instructions on how to use the tool. If you encounter any issues or have any questions when using PACE, please email PLOS at . PACE helps ensure that figures meet PLOS requirements. To use PACE, you must first register as a user. Registration is free. Then, login and navigate to the UPLOAD tab, where you will find detailed instructions on how to use the tool. If you encounter any issues or have any questions when using PACE, please email PLOS at figures@plos.org. Please note that Supporting Information files do not need this step.. Please note that Supporting Information files do not need this step.

---

## [Author Response · Author response to Decision Letter 2]

27 Dec 2025

We appreciate the Editor and Reviewers for their constructive and insightful comments, which have enhanced the quality and clarity of our manuscript. The delay in submitting the revised version is due to the time required to regenerate and acquire high-resolution figures, especially those generated by molecular dynamics simulation software, to ensure accuracy and presentation quality.

---

## [Decision Letter · Decision Letter 2]

19 Feb 2026

The Anti-Angiogenic and Anti-Proliferative Activity of an N-Benzoyl-3-phenyl Isoserine Derivative Containing 1,2,3-Triazole and Quinoline Rings

PONE-D-25-02987R2

Dear Dr. Hussein,

We’re pleased to inform you that your manuscript has been judged scientifically suitable for publication and will be formally accepted for publication once it meets all outstanding technical requirements.

An invoice will be generated when your article is formally accepted. Please note, if your institution has a publishing partnership with PLOS and your article meets the relevant criteria, all or part of your publication costs will be covered. Please make sure your user information is up-to-date by logging into Editorial Manager at Editorial Manager® and clicking the ‘Update My Information' link at the top of the page. For questions related to billing, please contact  and clicking the ‘Update My Information' link at the top of the page. For questions related to billing, please contact billing support..

Kind regards,

Wagdy M. Eldehna, Ph.d

Academic Editor

PLOS One

Additional Editor Comments (optional):

Reviewers' comments:

Reviewer's Responses to Questions

**Comments to the Author**

1. If the authors have adequately addressed your comments raised in a previous round of review and you feel that this manuscript is now acceptable for publication, you may indicate that here to bypass the “Comments to the Author” section, enter your conflict of interest statement in the “Confidential to Editor” section, and submit your "Accept" recommendation.

Reviewer #1: (No Response)

Reviewer #2: All comments have been addressed

2. Is the manuscript technically sound, and do the data support the conclusions?

Reviewer #1: (No Response)

Reviewer #2: Yes

3. Has the statistical analysis been performed appropriately and rigorously? 

Reviewer #1: (No Response)

Reviewer #2: Yes

4. Have the authors made all data underlying the findings in their manuscript fully available?

Reviewer #1: (No Response)

Reviewer #2: Yes

5. Is the manuscript presented in an intelligible fashion and written in standard English?

Reviewer #1: (No Response)

Reviewer #2: Yes

6. Review Comments to the Author

Reviewer #1: (No Response)

Reviewer #2: Thank you for your revisions. I have reviewed the updated manuscript and confirm that all necessary edits have been made. I therefore recommend that the article be accepted for publication.

7. PLOS authors have the option to publish the peer review history of their article (what does this mean?). If published, this will include your full peer review and any attached files.). If published, this will include your full peer review and any attached files.

.

Reviewer #1: No

Reviewer #2: **Yes:** Mahmoud S. ElkotamyMahmoud S. Elkotamy

---

## [Editor Report · Acceptance letter]

PONE-D-25-02987R2

PLOS One

Dear Dr. Hussein,

I'm pleased to inform you that your manuscript has been deemed suitable for publication in PLOS One. Congratulations! Your manuscript is now being handed over to our production team.

Kind regards,

on behalf of

Dr. Wagdy M. Eldehna

Academic Editor

PLOS One